# Study on the Antiviral Activities and Hemagglutinin-Based Molecular Mechanism of Novel Chlorogenin 3-*O*-β-Chacotrioside Derivatives against H5N1 Subtype Viruses

**DOI:** 10.3390/v12030304

**Published:** 2020-03-11

**Authors:** Wan-Zhen Shi, Ling-Zhi Jiang, Gao-Peng Song, Sheng Wang, Ping Xiong, Chang-Wen Ke

**Affiliations:** 1Department of Pharmaceutical Engineering, South China Agricultural University, Guangzhou 510640, China; shiwanzhen181@163.com (W.-Z.S.); songgp1021@scau.edu.cn (G.-P.S.); 2Shenzhen Key Laboratory of Marine Bioresource and Eco-Environmental Science, College of Life Sciences and Oceanography, Shenzhen University, Shenzhen 518060, China; jlz2004@szu.edu.cn; 3Key Laboratory of Molecular Biophysics of the Ministry of Education, College of Life Science and Technology, Huazhong University of Science and Technology, Wuhan 430074, China; shengwang@hust.edu.cn; 4Guangdong Provincial Center for Disease Control and Prevention, Guangzhou 511430, China; kecw1965@aliyun.com

**Keywords:** chlorogenin 3-*O*-β-chacotrioside derivatives, H5N1 subtype avian influenza virus, hemagglutinin, antiviral mechanism

## Abstract

The objective of this study was to investigate the inhibitory effect of chlorogenin 3-*O*-β-chacotrioside derivatives against H5N1 subtype of the highly pathogenic avian influenza (HPAI) viruses and its molecular mechanism. A series of novel small molecule pentacyclic triterpene derivatives were designed and synthesized and their antiviral activities on HPAI H5N1 viruses were detected. The results displayed that the derivatives UA-Nu-ph-5, XC-27-1 and XC-27-2 strongly inhibited wild-type A/Duck/Guangdong/212/2004 H5N1 viruses with the IC_50_ values of 15.59 ± 2.4 μM, 16.83 ± 1.45 μM, and 12.45 ± 2.27 μM, respectively, and had the selectivity index (SI) > 3, which was consistent with the efficacy against A/Thailand/kan353/2004 pseudo-typed viruses. Four dealt patterns were compared via PRNT. The prevention dealt pattern showed the strongest inhibitory effects than other patterns, suggesting that these derivatives act on the entry process at the early stages of H5N1 viral infection, providing protection for cells against infection. Further studies through hemagglutinin inhibition (HI) and neuraminidase inhibitory (NAI) assay confirmed that these derivatives inhibited H5N1 virus replication by interfering with the viral hemagglutinin function. The derivatives could recognize specifically HA protein with binding affinity constant KD values of 2.57 × 10^−4^ M and 3.67 × 10^−4^ M. In addition, through site-directed mutagenesis combined with a pseudovirion system, we identified that the high-affinity docking sites underlying interaction were closely associated with amino acid residues I391 and T395 of HA. However, the potential binding sites of the derivatives with HA did not locate at HA1 sialic acids receptor binding domain (RBD). Taken together, these study data manifested that chlorogenin 3-*O*-β-chacotrioside derivatives generated antiviral effect against HPAI H5N1 viruses by targeting the hemagglutinin fusion machinery.

## 1. Introduction

Highly pathogenic H5N1 avian influenza as a zoonotic disease was caused by A/H5N1 subtype virus in the orthomyxoviridae family, which is a severe infectious disease characterized by sudden death and exceptionally high mortality in poultry. Avian influenza A/H5N1 first emerged in domestic geese of southeastern China in 1996 [1,2], and the first human case of infection in 3-year-old children was confirmed in Hong Kong Special Administrative Region, China, in August 1997 [3]. Since then, H5N1 viruses have spread across East Asia, Europe, and West Africa, and have evolved into more than forty divergent clades and subclades [4,5,6,7]. Since the reemergence of H5N1 viruses in 2003, the reported outbreaks occurred successively in more than sixty countries and regions, leading to dramatic economic losses in the global poultry industries [7,8].

Furthermore, H5N1 virus strains are extremely infectious and can be transmitted from avian species to human through multiple pathways, although transmission of the H5N1 virus from birds to humans is inefficient. Unlike human influenza A viruses, H5N1 viruses can infect the lower respiratory tract, causing hypercytokinemia and increased tissue damage [9]. The mortality rate identified among infected individuals was greater than 60% reported to WHO [10]. Moreover, the hemagglutinin gene of HPAI A/H5 viruses were undergoing continuous evolution, generating emerging re-assorted subtypes and clades. So far, four clades 2.2.1.2, 2.3.2.1a, 2.3.2.1c, and 2.3.4.4 strains of H5N1 virus have become predominant pandemic strains with the continued evolution of HA gene [4,11,12,13,14,15,16]. It was reported that that H5N1 virus can acquire the capacity for airborne transmission between mammals without an intermediate host [17]. Undoubtedly, H5N1 viruses pose a serious threat to public health, as well as to the global economy. In view of a major public health concern, providing protection against severe infection and death caused by H5N1 has become a research focus in the zoonosis field.

Hemagglutinin (HA) is a crucial viral envelope spike glycoprotein, responsible for entry and infection of viruses into host cells via binding to sialic acid on the surface of target cells. The fusion peptide domain genome of HA has the most highly conserved sequences and is vital to the viral life cycle [18,19,20]. Thus, targeting at HA may be a highly effective influenza therapeutic strategy. It has been proved by the licensed hemagglutinin inhibitor Arbidol (umifenovir) [21,22,23]. In general, the HA inhibitors may interrupt viral attachment and the fusion process at the early infection stages, reduce viral load, and provide new combination therapeutics for the escape mutants. In recent years, research on H5N1 antiviral agents targeting at HA has attracted many attentions.

The previous research discovered that chlorogenin 3-*O*-β-chacotrioside XC-27 and chlorogenin 6-α-*O*-actyl-3-*O*-β-chacotrioside can effectively inhibit the envelope glycoprotein hemagglutinin of HPAI H5N1 subtype viruses [24]. Studies indicated that C_6_-OH replacement of aglycone moiety in XC-27 for the antiviral activity is crucial [25,26]. However, the molecular mechanism underlying HA inhibition by chlorogenin 3-*O*-β-chacotrioside derivatives remained unknown. This study aimed at elucidating the molecular interaction of these agents and HA binding sites via PRNT and hemagglutinin inhibition data, pseudovirion system combined with site-directed mutagenesis studies, and anti-viral against multiple H5N1 subtype strains as well as significant binding affinity. Our findings may be useful for further structural optimization of HA inhibitors.

## 2. Materials and Methods

### 2.1. Chlorogenin 3-O-β-Chacotrioside Derivatives Synthesized

The synthesis of target compounds XC-27-1, XC-27-2, UA-Nu-ph-5 was the same as described previously [24,25,26] (Figure 1).

### 2.2. Cell lines and Chemical Reagents

Madin-Darby Canine Kidney (MDCK) cells were obtained from ATCC (Manassas, VA, USA), LinX-A cells were donated by the Scripps Research Institute in San Diego, CA, USA. Competent cells DH5a and Stb13 (Trans Gen BiotechCo., Ltd., Beijing, China). Eight days old SPF chicken embryos (DaHuaNong Biotech Co., Ltd., Guangzhou, China). DMEM and 0.25% Trypsin-EDTA (Gibco, Grand Island, NY, USA), Purelink Plasmid Extraction Kit (Invitrogen, Carlsbad, CA, USA), penicillin-streptomycin antibiotic (Gbico, USA), DMSO (SIGMA-ALDRICH, USA), Luciferase Assay Kit (Promega, Madison, WI, USA), Cellular Lysate for Luciferase Assay (Promega, USA), Lipofectamine^TM^ 3000 Transfection Reagent (Promega, USA), high fidelity enzyme and Dpn I enzyme (Takara, USA), Neuraminidase inhibitor screening reagent (Beyotime Biotech Co., Ltd., Shanghai, China), H5 antibody and H5 antigen (Harbin veterinary research institute, Harbin, China), HA protein (Sino Biological Inc., Beijing, China), Biotin Labeling Kit PN21343 (Thermo Scientific Pierce Protein Biology, USA), Agarose (Invitrogen, USA), Crystal Violet (Damao Chemical Reagent Factory, Tianjin, China).

### 2.3. Cytotoxicity Evaluation of Compounds

Cytotoxicity effect of compounds was evaluated by MTT assay. Briefly, MDCK and LinX-A cells (10^5^/mL) were seeded in T25 flasks with DMEM containing 10% FBS and 1% penicillin- streptomycin in a final volume of 6 mL and then incubated at 37 °C in a humidified air with 5% CO_2_ (Thermo-Fisher, USA) for 48 h. When the cells reached 80%–90% confluence, the culture medium was discarded, and the cells were washed twice with PBS, then cells were digested with 2 mL 0.25% Trypsin-EDTA solution at 37 °C for 3–5 min. After that, harvest cells with centrifugation at 300× g for 5 min. Once again, cells were resuspended in 5 mL complete media and inoculated in 96-well culture plates at the density of 1 × 10^4^/well (MOXI^TM^ cell counter, ORFLO Technologies, USA) in a final volume of 0.1 mL. Cells were incubated at 37 °C for at least 24 h to allow optimal attachment. When the cells reached 60% confluence, they were treated with 0.2 mL volume of the tested compounds in a specific concentration, and incubated at 37 °C for 48 h. Medium were exchanged once every 24 h. Cell control group and blank group were set up in the same manner, with six parallel wells for each group. Cell survival was assessed by directly adding 22 µL of 5 mg/mL MTT to 0.2 mL medium. After four hours, the formazan precipitate was dissolved in 150 µL DMSO per well. Afterwards, the optical density of each well was detected at 570 nm wavelength by GENiosPro microplate reader (TECAN, Switzerland). The CC_50_ values were calculated according to average OD values by applying Prism 5.0 software. This experiment was repeated at least three times.

### 2.4. Titration of Influenza Viruses and TCID_50_ Assay

A/Duck/Guangdong/212/2004 (H5N1) virus strain was amplified by SPF chicken embryos in biosafety cabinet class III (ESCO, Singapore) of P3 laboratory and stored at −80 °C. MDCK cells were seeded in 96-well plates at the density of 1 × 10^4^/well with DMEM containing 10% FBS and 1% penicillin-streptomycin in a final volume of 0.2 mL. Cells were incubated at 37 °C in a 5% CO_2_ incubator for at least 24 h to allow optimal attachment. Cell control group and blank group were set up in the same manner, with at least six parallel wells for each group. After 24 h, cells were overlaid with virus 10-fold serially diluted in PBS with 1% FBS and incubated for 1 h at 37 °C.

The inoculum was removed, and the cells were washed twice with PBS and supplied with fresh DMEM containing 0.2% bovine serum albumin and trypsin (5 µg/mL). The plate was incubated for 48 h at 37 °C. On the fourth day after infection, the cytopathic effect (CPE) was observed with a microscope (Nikon, Japan), and H5N1virus titer was determined by TCID_50_ assay and calculated by the Reed–Muench method.

### 2.5. Determination of IC_50_ for Anti-Influenza Virus Activity

IC_50_ value of the tested compound against influenza viruses was evaluated by MTT assay. MDCK cells were inoculated at a density of 1 × 10^5^ cells/mL in a 96-well plate with 100 µL per well. Cells were incubated at 37 °C for 24 h to allow optimal attachment. Cell control group and blank group were set up, with at least six parallel wells for each group. The tested compounds were diluted by a serial 10-fold dilution way. The diluted compound and 100TCID_50_ virus were co-incubated in a 5% CO_2_ incubator at 37 °C for 30 min, then they were added to MDCK cells and continued to incubate for 1 h. After 1 h, cells were washed twice with PBS and incubated with bovine serum-free DMEM at 37 °C for 48 h. On the fourth day after infection, cell viability was measured by MTT assay. The optical density of each well was detected at 570 nm wavelength by GENiosPro microplate reader (TECAN, Switzerland), and IC_50_ values of the tested compounds were calculated using GraphPad prism 5.0 (GraphPad Software, La Jolla, CA, USA). This experiment was repeated at least three times.

### 2.6. Plaque Reduction Neutralization Test (PRNT)

MDCK cells were grown in 6-well plates to confluence and infected with H5N1 virus (100 TCID_50_) for 1 h at 37 °C, compound or ribavirin was added before, during, or after H5N1 infection, the virus micro-plaques assay was performed. In this study, vehicle control group and ribavirin group were set up. The compounds were performed two-fold serial dilution with DMEM to obtain diluted solutions of 1:10 to 1:160. Four infection patterns for cells were taken. For prevention, cells were incubated with 500 µL derivatives at various concentrations for 2 h at 37 °C, and then medium was replaced with 500 µL of H5N1 viruses and incubated for 1 h. For inactivation, 500 µL of H5N1 viruses were first incubated with 500 µL derivatives for 30 min before added to cells. For neutralization, cells were simultaneously incubated with H5N1 viruses and the derivatives [27,28,29]. After the removal of mixture, each well was overlaid with 2 mL maintenance medium containing 1% of agarose. For treatment, cells were first infected with 500 µL of H5N1 viruses for 1 h, followed by two washes with PBS and each well was overlaid with 2 mL medium containing the derivatives and 1% of agarose. After that, the 6-well plates were inverted in a 5% CO_2_ incubator and incubated at 37 °C. After 48 h, the medium was removed and cells were fixed with 500 µL of 4% paraformaldehyde solution for 1 h at room temperature. Subsequently, the fixed cells were stained with 600 µL of 0.2% crystal violet solutions at 37 °C for 4 h. In the end, the solutions were discarded, following by four times washes with ddH_2_O_2_ and allowed to dry. Finally, plaque formations were observed with an inverted microscope (Nikon, Japan) at an original magnification of 10 × and the plaque formation were counted. This experiment was repeated at least three times. The plaque-forming units (PFUs) were calculated using the following formula: PFUs/mL= (average number of plaques per well/the inoculated virus amounts per well (mL)) × the degree of virus dilution.

### 2.7. Hemagglutinin Inhibition Assay (HI Test)

Prepare 1% (*v*/*v*) chicken RBCs suspension. The 4 HA units of the virus solution were determined. The hemagglutinin inhibition effects of compounds were determined by the HI test. Firstly, 25 µL PBS was pipetted into 96-well microtiter plate. Then, 25 µL of the tested compound solution was added to each well of the 1st column. A serial two-fold dilution with 25 µL PBS was performed from the 2nd column to the 10th column, and 25 µL mixture of the last well was discarded. The wells of the 11th column served as the erythrocyte negative control which was filled with 25 µL PBS. The wells of the 12th column served as the positive control which was filled with 25 µL H5 antigen. Subsequently, 25 µL of the prepared virus solutions with 4 HA units were added to each well. Finally, 25 µL of 1% (*v*/*v*) chicken RBCs suspension was added to each well. Each well was fully mixed by shaking for 1 min. The tested derivatives were incubated with the virus solutions with 4 HA units for 60 min at 4 °C. Afterwards, the hemagglutinin inhibitory results were analyzed. The experiment was repeated at least three times.

### 2.8. Assay of Neuraminidase Inhibitory Activities

Inhibitory rates of H5N1 virus neuraminidase activity was determined by FL-MU-NANA method [30] (4-MU-NANA; Sigma, St. Louis, MO, USA). The compounds were dissolved in DMSO to gain a 4 mM concentration solution, and then the solution was two-fold serial diluted with ddH_2_O to a final concentration of 2, 1, 0.5, 0.25, 0.125 to 0.0625 mM. Standard curves were prepared using neuraminidase inhibitor screening kit according to the manufacturer’s instructions. Briefly, 70 µL of reaction buffer solutions were added to each well, and then added NA enzyme with 0, 1, 2, 5, 7.5, and 10 µL volume to the corresponding well. In the end, each well was supplemented with ddH_2_O to a final volume of 90 µL. Vibration mixing was performed for 1 min and incubation for 2 min at 37 °C. Afterwards, 10 µL of the fluorescence substrate was added to reach a total of 100 µL reaction mixture. After shaking for 1 min, the plate was sealed and incubated for 30 min at 37 °C. Finally, the fluorescence was read using a spectrophotometer (Hitachi, Japan). In the same way, a reaction mixture containing 70 µL of reaction buffer solution, 10µL of NA enzyme, and the different compound dilutions with volume of 10 µL were added to the wells respectively. The rest was as described previously. Each sample had at least six parallel wells. The inhibition (%) was calculated using the formula:NA Inhibitory activity (%) = [1 − (F_S_ − F_0_)/ (F_C_ − F_0_)] × 100%.(1)

Among them, fluorescence intensity (F) was quantified with excitation wavelength at 360 nm and emission wavelength at 450 nm. F_S_ was fluorescence intensity of inhibitors in the presence of the sample, F_0_ was the fluorescence intensity of the blank control, and F_C_ was the fluorescence intensity of the negative control. The 50% inhibitory concentration (IC_50_) was analyzed by probit regression in SPSS (version 21, IBM SPSS, Chicago, IL, USA). IC_50_ values represented the mean of three individual determinations, each performed in triplicate assays.

### 2.9. Plasmid Construction

HIV backbone vectors, pNL4-3.luc.R-E- and VSVG plasmid (AIDS Research and Reference Reagent Program, Division of AIDS, NIAID, NIH), A/Thailand/Kan353/2004 (H5N1) HA, and NA plasmids (Pharmacy College of Southern Medical University) were amplified by PCR and further verified by sequencing. Eventually, the infectious pseudo-typed viruses were generated through co-transfection of pNL4-3.luc.R-E- or VSVG vectors with env expression plasmids NA and HA to cells. Cells can be infected one time in virtually any format. The gene sequences of constructed plasmids were verified by the sequence analysis. The specific primers were: A-H5N1-HA: 5′-ATATTTCCGTTGGGACATCAACACTA-3′ (Forward), 5′-CCAAAGTAAACGGGCAAAGTGGA

A-3′ (Reverse). The DNA fragment assembly used ChromasPro software. DNA sequence of HA plasmid was compared with original sequences of the NCBI database using DNAssist software (Appendix A).

### 2.10. Preparation of H5N1 and VSVG Pseudovirus

H5N1 pseudoviruses were prepared with three plasmids by co-transfection methods using the calcium phosphate technique [31], which included 0.5 μg ENV expression plasmid HA, NA and 1.5 µg pNL4-3.Luc.R-E- (Lipofectamine^TM^ 3000 Transfection Systems, Promega, USA). Likewise, VSVG pseudoviruses were prepared with VSVG plasmid. Briefly, LinX-A cells were inoculated in 6-well culture plates at a density of 3 × 10^5^/mL cells in a final volume of 2 mL. Cells were incubated at 37 °C for 24 h. When the growth of cells reached 80% confluence, the medium was replaced with medium containing 1 mM sodium butyrate (Sigma-Aldrich), transfection process was performed using Lipofectamine^TM^ 3000 transfection reagent kit according to the manufacturer’s instructions.

Firstly, to aspirate 3.75 µL of transfection reagent into a sterile EP tube, and adjusted to a final volume of 125 μL with DMEM, and then fully mix. Aspirated 1.5 µg of pNL4-3.Luc.R-E-, 0.5 μg of HA plasmid, 0.5 μg of NA plasmid, and 5 μL of P3000^TM^ reagent into another sterile EP tube, and adjusted to a final volume of 125 μL with DMEM. Finally, the two solutions were mixed at 1:1 ratio and stood at room temperature for 15 min. Cells in 6-well plates were incubated with this mixture at 37 °C for 48 h. Two days after transfection, the supernatants were collected by centrifugation at 300× *g* for 5 min (Millipore NV, Brussels, Belgium), the pseudotyped viruses were harvested. In the end, the prepared pseudo-typed viruses were sub-packaged and stored at −80 °C for use.

### 2.11. The Infectivity Ability Assay of Pseudovirus

MDCK cells were seeded in a 96-well plate at the density of 1×10^4^/well in a final volume of 0.2 mL. After cells were incubated at 37 °C for 24 h, the medium was replaced with a two-fold diluted H5N1 pseudovirus solution at 0.2 mL volume of each well to infect MDCK cells. Cell control group and blank group were set up with at least six parallel wells per group. After 48 h, the medium was discarded and cells were washed twice with PBS. The infectivity ability of pseudovirus was measured using Luciferase Assay Kit (Promega, USA) according to the manufacturer’s instructions. Briefly, the cells were lysed with 50 µL of cell lysing reagent per well, then to aspirate 40 µL cell lysis solutions from each well to a 96-well luciferase assay plate. Subsequently, 40 μL of luciferase substrate was added into each well in dark conditions. Immediately, the luciferase chemiluminescence value was assayed using a microplate reader (Tecan, USA). The infectivity ability of pseudoviruses was determined according to the relative light intensity value. In this experiment, mock controls were included, which consisted of VSVG and pNL4-3.Luc.R-E-. The experiments were repeated independently at least three times.

### 2.12. Inhibition Effects of Compounds against H5N1 Pseudovirus

MDCK cells were seeded in a 96-well plate at the density of 1 × 10^4^/well in a final volume of 0.2 mL at 37 °C for overnight. The tested compounds were performed two-fold serial dilution with DMEM to obtain different concentrations. Subsequently, 50 µL of the diluted compound solution was added into 50 µL of H5N1 pseudoviruses and co-incubated for 30 min at 37 °C. Finally, they were added into MDCK cells and continued to incubate for 48 h. In the same way, vehicle control group and negative control group were set up with six parallel wells for each group. After 48 h, the medium of each well was discarded and cells were washed twice with PBS. The infectivity ability of pseudoviruses was measured using Luciferase Assay Kit (Promega, USA) according to the manufacturer’s instructions. The operation approach was as previous described. The susceptibility of pseudoviruses to the tested compounds was determined according to the chemiluminescence value. The inhibition rates (%) of compounds for viruses were calculated using the formula:Inhibitory rates (%) = [1 − (E − N)/(P − N)] × 100%.(2)

Among them, E represented the fluorescence intensity of samples in the presence of the pseudoviruses, P represented the fluorescence intensity of only pseudoviruses, and N was the fluorescence intensity of the negative control.

### 2.13. Prediction of Molecular Docking for Compounds

Molecular docking is an effective approach for studying molecular interactions. To better understand the most likely space conformation of interaction between HA and the compounds, and predict the possible molecular docking sites, we carried out the computational structural simulation analysis. In this study, the AutoDock Software package with graphics interface, AutoGrid/Auto Dock version 4.2.6 and Vina were applied to docking procedures. Crystal structure of VN1194 (H5N1) HA with PDB ID: 2IBX was obtained from the RCSB protein databank (RCSB-PDB). The structure conversion and preprocess were conducted, all water molecules were removed, and polar hydrogen atoms and charges were added to the refined model using AutoDock Tools. Some key amino acid residues located in the potential active domain of HA were subjected to a flexibility process. The constructed protein structure was saved in PDBQT format. After that, a rectangular box was defined for configuration of the binding site. AutoGrid was used for the preparation of the grid map. AutoDock/Vina was employed for docking according to the information of HA and the tested compounds along with grid box properties in the configuration file. During the course of docking, structures of HA protein and the compounds were considered as rigid. The pose with lowest energy of binding or binding affinity was extracted and aligned with the receptor protein structure for further analysis. The Lamarckian Genetic Algorithm (LGA) was chosen to search for the best conformers.

### 2.14. Site-Directed Mutagenesis of the Predicted Binding Sites

Based on the predicted docking sites, site-directed mutations were introduced into HA via PCR (Bio-Rad, USA) by designed primer according to HA sequence of A/Thailand/kan353/2004 virus. The site mutation on HA was generated using the Stratagene QuikChange multi-site-directed mutagenesis system. The mutant primers were shown in Appendix A. PCR amplification was performed according to the manufacturer’s instructions. The PCR products were subjected to restriction endonuclease reaction. The enzyme-digested products were transformed into *E. coli* cells for further colony PCR screening. Extract DNA plasmids using the QIAprep spin miniprep kit (QIAGEN, Germany) for DNA sequencing. The sequencing primers were as follows: HA-48SQ-R, 3′-GGCTTCATGACTGGACCAAG-5′; HA-(329,331) SQ-F, 5′-GTACCAAGAATAGCTACTAGATC-3′; HA-(367,387,391, 394, 395) SQ-F, 5′-TGGAATATGGTAACTGCAACAC-3′. Pick the correct clones for glycerol stock and store at −80 °C for use.

### 2.15. Inhibition Effects of Compounds against Site-Mutant H5N1 Pseudovirus

To determinate whether the interaction of HA and derivatives were involved in amino acid residues I48, L329, T331, W367, T387, I391, V394, and T395 in HA, we further compared the susceptibility of wild-type pseudoviruses to the derivatives with that of the site-mutant H5N1 pseudoviruses. In the study, site-mutant A/Thailand/kan353/2004 H5N1 pseudoviruses were prepared as described previously. On the basis of the infectivity ability, the inhibitory rates of derivatives against site-mutant H5N1 pseudoviruses were determined according to the chemiluminescence value. The inhibition rates (%) of compounds were calculated using the following formula: Inhibitory rates (%) = [1 − (E − N) / (P − N)] × 100%. Among them, alphabets mean the same as above.

### 2.16. Bio-Lay Interferometry (BLI) for the Binding Ability Analysis

The binding ability for HA and derivative was analyzed by BLI assay, which includes procedures of equilibrium, immobilization of the ligand, association and dissociation of compounds. Before detection, the sensor was pre-balanced with PBST buffer so as to minimize nonspecific binding. The super-streptavidin (SSA)-coated biosensor tips were infiltrated in PBST for 10 min to remove their protective sucrose coat and achieve a sufficient baseline signal. The biotinylated HA protein (Sino Biological Inc., China) was prepared using the biotin label kit according to the manufacturer’s instructions. Briefly, 200 μL of 20 μg/mL HA was mixed to 200 μL of biotin. The mixtures were incubated at room temperature for 30 min. Subsequently, the mixtures were desalted by Spin column (Thermo Scientific, USA). Dilute the biotinylated HA with PBST buffer to two-fold serial dilutions. Afterwards, the diluted biotinylated HA was loaded onto the SSA biosensor tips and incubated for 5 min, followed by another equilibration. The balanced SSA-coated biosensor tips with biotinylated HA were immersed into the two-fold serial diluted compounds in black 96-well plates and carried out association for 45 s. Finally, the compounds were dissociated from the biotinylated proteins for 45 s. This experiment was repeated at least three times. In this assay, ForteBio Data Acquistion software (Pall-Fortebio, USA) was applied to collect the real-time data on the molecular interaction and binding kinetics. The results were fitted to kinetic equation using steady state analysis method and calculated the intermolecular affinity (K_D_), the association rate (K_a_), and dissociation rate (K_d_) and so on.

### 2.17. Statistical Analyses

The SPSS statistical Software package (SPSS Inc. Chicago, IL, USA) with Windows Version 20.0 was applied for statistical analyses of the experiment data. The statistical results are presented as the mean ± standard deviation (SD). The comparison between groups was conducted using analysis of one-way variance (ANOVA). If the variance is homogeneous, the LSD test for multiple comparisons should be followed by that analysis, if not, Tamhane’s T2 (M) or Dunnett’s T3 test were used. Levels of significance were set at *p* <0.05, *p* < 0.05 indicated that the difference was statistically significant, and *p* < 0.01 indicated that the statistical difference was extremely significant. GraphPad Prism 5 software (GraphPad Software, Inc, San Diego, CA, USA) was used for plotting. For every experiment, data were representative of at least three independent experiments.

## 3. Result

### 3.1. Determination of Infectivity Titration

Based on the observed cytopathic effect (CPE), the titer of infectious H5N1 viral progeny was determined in terms of TCID_50_ per milliliter using the Reed–Muench method, which provided viral titer for further detection of antiviral activity assay and PRNT. Anyway, according to the results (Appendix A), the TCID_50_ of A/Duck/Guangdong/212/2004 (H5N1) virus was 10^−6.5^/0.1 mL.

### 3.2. Antiviral Activity of the Compounds against H5N1 Viruses

To estimate the safety profile, the vitro cytotoxicity study of compounds was performed. In the present study, we used MTT assay to investigate the cytotoxicity of these compounds on MDCK cells. The half-cytotoxicity concentration (CC50) values were calculated according to the inhibitory percentage of the tested compounds at various concentrations on the viabilities of cells (Table 1). Comparison of these tested compounds, UA-Me-5 was observed to have strongest cytotoxic effects on the growth of MDCK cells than other compounds. It was apparent that the chlorogenin 3-*O*-β-chacotrioside derivatives XC-27-1 showed the lowest cytotoxicity with CC_50_ value 1247 ± 1.16 μM. Actually, the vitro cytotoxicity of XC-27-1 was significantly lower than the toxicity of ribavirin, and also decreased by 16-fold than that of XC-27-2.

We further investigated the half-inhibitory concentration (IC_50_) values and the selective index (SI) of the synthesized compounds. SI was calculated as the ratio of the mean of CC_50_ value to the mean of IC_50_ value. The criteria for determining compound activity are based on its SI. Compounds with an SI value of >3 were defined as active, whereas compounds that exhibited an SI value less than 3 were defined as inactive. In the present study, ribavirin was selected as a positive control drug. As shown in Table 1, among the synthesized compounds, UA-Nu-ph-5, XC-27-1 and XC-27-2 had their SI values more than 6, it was obvious that these three derivatives displayed low toxicity and better inhibitory activity against the HPAI H5N1 virus than the others. The inhibitory activity of XC-27-1 showed highly selective in particular.

As shown in Figure 2 and Appendix A, the results indicated that the derivatives UA-Nu-ph-5, XC-27-1, and XC-27-2 inhibited H5N1 avian influenza virus in a concentration-dependent manner, that is, the inhibition percentage against H5N1 viruses significantly increased with the log concentration increasing of derivatives. The IC_50_ values of UA-Nu-ph-5, XC-27-1 and XC-27-2 respectively presented 15.59 ± 2.4 μM, 16.83 ± 2.27 μM, and 12.45 ± 2.27 μM. Among them, XC-27-2 showed the strongest inhibitory activity against H5N1 virus than the other two derivatives.

### 3.3. H5N1 Pseudovirion Infection of MDCK Cells Assay

The infectivity of the prepared pseudovirions for cells was evaluated by the luciferase activity assay. The HA gene sequences of wild-type A/Thailand/kan353/2004(H5N1) virus were retrieved from NCBI databases. The nucleotides IDs are EF541411.1. The corresponding amino acid IDs are ABP51985.1. The gene sequences of HA plasmid were aligned with that of A/Thailand/Kan353/2004(H5N1) virus strain. Analysis results indicated that HA plasmid had a mutation at residue 300 within HA, that is, base C substituted base T (Appendix A). However, there was not any change in the corresponding amino acid. If the prepared pseudoviruses successfully infected cells, the luciferase reporter gene of pNL4-3.Luc.R-E- plasmid would be expressed within the cells. Therefore, the infectivity of the pseudoviruses can be reflected by the fluorescence intensity value assay. In the assay, the blank control group had no treatment, and 2.5 folds of their light intensity value were defined as a cutoff value. As defined by the manufacturer’s instructions, the pseudoviruses showing ‘highly infectivity’ are that their fluorescence intensity have a > 100-fold cutoff value in 10-fold diluted concentration. The results showed that the H5N1 pseudovirus 10-fold diluted had a >10^5^–fold cutoff value, actually its average fluorescence intensity value equaled 26489122, which exceeded 10^7^ RLU (Appendix A and Appendix A). The VSVG pseudovirus 10-fold diluted also had a > 10^3^-fold cutoff value. The results indicated the two prepared pseudoviruses showed highly infectivity to MDCK cells.

### 3.4. H5N1 Pseudovirion-Based Hemagglutinin Inhibition Assay

The envelope glycoprotein G from vesicular stomatitis virus (VSVG) has extremely broad tropism, and can be transduced efficiently and irreversibly into many cell types genome. So VSVG was widely used for the pseudotyped retroviral vector. In this study, VSVG-pseudotyped viruses were used as a vehicle control, which was prepared through the VSVG gene cloned into the pNL4-3.Luc.R-E- vector. Additionally, H5N1-pseudotyped viruses were constructed through the envelope glycoprotein HA and NA gene cloned into the pNL4-3.Luc.R-E- vector. These pseudotyped viruses possess single-round replicative competence. The relative light unit (RLU) values of pseudotyped viruses present positive correlation with their infectivity. When the derivatives block the envelope glycoproteins HA and NA, the viral infectivity and the luciferase activity RLU value will decrease. So, the RLU value changes can reflect the susceptibility of H5N1 pseudotyped viruses to the derivatives.

As shown in Appendix A and Figure 3, the infection of A/Thailand/kan353/2004 (H5N1) pseudo-typed virus was strongly inhibited by XC-27-1 and XC-27-2. Actually, XC-27-1 and XC-27-2 presented inhibitory effects against the H5N1 pseudovirus in a concentration-dependent manner and with the IC_50_ value 12.76 ± 7.01 μM, 18.47 ± 1.82 μM, respectively. However, they had no inhibitory effects on the VSVG pseudovirus. Therefore, the results indicated that the derivatives can interrupt with NA and HA.

To further identify whether the derivatives target at NA, NAI susceptibility assay was performed. Firstly, the standard curve was set up. Then, the fluorescence-based neuraminidase inhibition assay was conducted using fluorescence substrate in accordance with the manufacturer’s instructions to survey NA activity after treated with the derivatives. As shown in Figure 4 and Appendix A, although at a high concentration of 400 μM, XC-27-1, and XC-27-2 showed very low inhibition activity on NA.

Compared with the NA inhibitory activity, the antiviral activity of the derivatives against H5N1 pseudoviruses was far more beyond than that. Anyway, the results indicated that the derivatives inhibited the infection of H5N1 viruses by targeting at HA.

### 3.5. Analysis of H5N1 Virus Entry Inhibition with PRNT

To further confirm the antiviral activity of chlorogenin 3-*O*-β-chacotrioside derivatives against the H5N1 virus in vitro, and identify the potential inhibitory stages, PRNT was performed on the basis of the observed CPE and the PFU counts.

As shown in Appendix A and Figure 5a, the prevention pattern brought XC-27-1 the strongest inhibitory effects against H5N1 viruses via observing CPE and PFUs, and the inhibition rate of viral plaques forming reached 95% when treated with 40 μM XC-27-1. Under prevention and neutralization patterns, XC-27-1 also had significant inhibitory effects on plaques forming amounts. Furthermore, the inhibitory efficacy of the three patterns presented a dose–response manner. However, XC-27-1 showed lowest inhibitory activities after H5N1 virus entry (treatment). In contrast to XC-27-1, ribavirin more strongly reduced the viral plaques forming amounts after virus entry rather than before virus entry. As a result, XC-27-1 protected MDCK cells from HPAI H5N1 virus infection via interfering with the early stage of virus life cycle, and it had a different antiviral mechanism from ribavirin. The same as to XC-27-1, XC-27-2 also showed stronger inhibition effect on plaques forming amounts before H5N1 virus entry rather than after virus entry (Appendix A and Figure 5b), and the plaque reduction presented a dose-dependent manner. In fact, for prevention and neutralization, the inhibition percentage of H5N1 viral plaques forming amounts reached 96% and 91%, respectively at 40 μM concentration. Anyway, UA-Nu-ph-5 also showed the same inhibition effects on plaques forming amounts as XC-27-1 (Appendix A and Figure 5c).

On the whole, the study results verified that three derivatives XC-27-1, XC-27-2, and UA-Nu-ph-5 had significant inhibitory effects on the HPAI H5N1 virus before virus entry. It suggested that the derivatives prevented the entry process at the early stages of the H5N1 viral infection by targeting HA attachment or fusion machinery. Regarding the reason that the three derivatives showed highest effect when the derivatives had contact with cells before virus entry (prevention), it was possible that chlorogenin 3-*O*-β-chacotrioside derivatives may produce antiviral effects through other cell signal pathways in addition to target at HA protein [32]. 

### 3.6. The Binding Site Analysis of HA1 and Compounds

The surface antigen (H5) of the H5N1 virus is located at the HA head (HA1 subunit). When H5 antigen binds to the sialic acid receptor of chicken RBC, hemagglutination reaction will happen. Thereby, through the hemagglutination inhibition reaction, we can judge whether the derivatives block antigen sites of HA1. As defined by the criteria, 100% of RBCs was agglutinated, which displayed ‘100% evenly spread’ on the bottom of the wells, the ‘++++’ symbol was used. When hemagglutination reaction displayed almost 100% evenly spread of RBCs with a large circle around, the ‘+++’ symbol was used. If hemagglutination reaction displayed a medium circle with small clot dots around, the ‘++’ symbol was used. Only a small circle with a few colt dots around was recorded as the ‘+’ symbol. The ‘−’ symbol was used to record the absence of hemagglutination.

As shown in Figure 6, for the vehicle control group, no hemagglutination reaction was generated. On the contrary, the H5-antigen group had a hemagglutination reaction. While the reaction no longer was generated with the increase of the dilution ratio. Anyway, according to the results, the hemagglutination titer was a 2^5^–2^6^-fold diluted concentration of the H5-antigen.

Each dimer of the HA has two units, it is divided into HA1 and HA2 when hydrolyzed. HA1 possesses the sialic acid receptor binding site (RBD), viruses can bind to the susceptible target cells through RBD. HA2 mediates subsequent entry via fusion of the viral membrane with a host cell membrane.

The hemagglutination inhibition assay mainly detects whether HA1 protein was the potential target of derivatives. As shown in Figure 7, the same as the H5 antigen group, XC-27-1 and XC-27-2 had not produced inhibitory effects to the hemagglutination reaction induced by the virus HA1. This suggested that the derivatives cannot block the attachment action of the viruses with target cells via targeting RBD sites of H5N1 viruses, indicating that the derivatives may possibly interrupt with HA2 fusion via binding to the HA2.

### 3.7. The Binding Affinity Analysis of HA Protein and Compounds

In order to confirm and characterize the interaction, we further studied direct binding affinity and real-time rates of association and dissociation between derivatives and HA. In general, kinetics determines whether a complex forms or dissociates within a given time span, and equilibrium dissociation constant (K_D_) can report molecular binding affinity. Anyway, K_D_ value, as a ratio of the dissociation constant (K_d_) to the association constant (K_a_), is a commonly used parameter to evaluate the strength of molecules interaction. K_D_ has nothing to do with compound concentration. The smaller the K_D_ value is, the greater the binding affinity of the compound to HA would become.

BLI can generate quantitative and real-time kinetic parameters. In this study, we applied bio-layer interferometry technology (BLI) to analyze the interaction, and obtained K_D_ values from the Octet RED system workstation of ForteBio Octet K2 instrument (Pall-Fortebio, USA). As shown in Appendix A, five curves represent the association–dissociation curves of derivative binding to HA. Actually, we obtained the response values of the derivatives at varying concentrations (125, 62.5, 31.3, 15.6, 3.91 μM). The study results indicated that the signal intensity of the interaction between binding partners became strengthened when the concentration of the derivative solution was increased. The interaction between HA and derivatives presented significant dosage–efficacy relationships. The results indicated that the derivatives XC-27-1 and XC-27-2 can recognize specifically and bind to HA. In addition, the binding affinity K_D_ was calculated according to the curves of association and dissociation of XC-27-1 and XC-27-2, that is, the corresponding K_D_ values equaled to 2.57 × 10^−4^ M and 3.67×10^−4^ M, respectively (Table 2).

### 3.8. Specific Binding Sites Analysis of Compounds to HA

As shown in Figure 8, computational simulations showed that derivative XC-27 generated interaction with high fidelity stem domain of HA protein (Figure 8), and potential molecular docking sites lay mainly on the amino acid residues I48, L329, T331, W367, T387, I391, V394, T395.

To further verify whether eight amino acid residues play major roles on the process of interaction, base substitution was introduced into HA by site-mutation technique. In addition, the infectivity of mutant H5N1 pseudo-typed viruses to MDCK cells was assayed using A/Thailand/kan353/2004 H5N1 pseudoviruses as a reference virus strain. As defined by the instructions, pseudoviruses with 10-fold diluted have a > 100-fold cutoff value, that are considered highly infective. The results indicated that the RLU of all mutant pseudoviruses had significant difference from that of wildtype H5N1 pseudoviruses except for the HA-L329F mutant strain (*p* > 0.05). The RLU of HA-I48A and HA-W367R mutant strains had no significant difference when compared to that of the vehicle control group (*p* > 0.05) (Figure 9 and Table 3). Anyway, the mutant virus strains with residues I48A and W367R substituted in the HA had no infectivity. Thereby, it suggested that residues sites I48 and W367 of HA may be closely related to the HA function. However, the rest site-mutant pseudoviruses can infect MDCK cells, although their infectivity slightly reduced.

After H5N1 subtype pseudoviruses infected MDCK cells, the antiviral activities of the derivatives XC-27-1 and XC-27-2 were subjected to the IC_50_ assay. In this work, we tested the antiviral activities of the derivatives to mutant pseudotyped viruses by the luciferase assay method under the gradient concentration of 1.56, 3.13, 6.25, 12.5, 25, 50, 100, and 200 µM, respectively, and then the IC50 value was calculated according to these data. All of the H5N1 pseudotyped viruses with residues L329F, T331A, T387A, I391A, V394A, and T395A substituted in HA gained infectivity to MDCK cells except for pseudo-typed viruses with HA-I48A and HA-W367R mutation.

In this study, we conducted a comparative analysis on the susceptibility of the derivatives against wild-type A/Thailand/kan353/2004(H5N1) pseudotyped viruses versus the mutant H5N1 pseudotyped viruses. As shown in Figure 10a and Table 4, XC-27-1 remained significant inhibitory effects against mutant H5N1 pseudo-typed viruses which had substituted in L329F, T331A, T387A, and V394A within HA. The IC_50_ values of XC-27-1 against mutant viruses with HA-T331A were substituted and wildtype viruses were less than 12.76 µM. However, the susceptibility of mutant H5N1 pseudovirus with HA-I391A or HA-T395A substituted to XC-27-1 significantly decreased, with the dose–respond curves declining and shifting to the right, this represented that the generated affinity between XC-27-1 and HA declined, with the IC_50_ values 67.20 ± 8.27 μM and 48.34 ± 10.18 μM, respectively. In addition, the derivative XC-27-2 also showed good susceptibility to H5N1 pseudo-typed viruses with L329F, T331A, T387A, V394A, and T395A mutation in HA (Figure 10b and Table 5). The IC_50_ values of XC-27-2 against mutant viruses with L329F and T331A substituted and wildtype viruses were less than 18.47 µM. The same as that of XC-27-1, the susceptibility of pseudo-typed viruses to XC-27-2 also declined when residue I391A substituted. Meanwhile, the dose–respond curves of XC-27-1 declined and shifted to the right. It signified that the affinity of XC-27-2 binding to HA became a significant reduction. Actually, the IC_50_ value of XC-27-2 against virus enhanced from 18.47 ± 1.82 μM to 95.29 ± 7.71 μM. To sum up, we draw a conclusion that XC-27-1 most likely generated interactions with residues I391 and T395 of HA, and XC-27-2 also had an interaction with residues I391 of HA. 

## 4. Discussion

Highly pathogenic avian influenza H5N1 virus, a single-stranded segmented negative-sense RNA virus, remains a pandemic threat. Since the first case of human infection was confirmed in August 1997 [3], HPAI H5N1 viruses have continued to cause severe human disease in countries where poultry are infected [10,33]. In addition, the variation speed of the H5N1 virus was amazing. In 2005, drug-resistant H5N1 virus strains were reported [34]. Currently, M_2_ ion channel inhibitors, such as amantadine and rimantadine, lose inhibition activity to H5N1 viruses due to antiviral resistance. As a result, neuraminidase inhibitors (NAIs) became recommended antiviral drugs of high-risk H5N1 disease. However, H5N1 viruses continually evolve under immune pressure, antigenic drift, and genetic re-assortment, which had caused the susceptibility of H5N1 viruses to NAIs drugs to decrease, even antiviral resistance [35,36,37]. Thereby, it is urgently required to develop new antiviral agents with novel action mechanisms to prevent/control the pandemic spread of NAIs-resistant strains.

Influenza HA is a viral membrane glycoprotein, which is responsible for entry and infection through binding to terminal sialic acids on cellular receptors and mediating fusion. The fusion peptide domain of HA has the most highly conserved sequences, and it is vital to the viral life cycle, pathogenicity, and virulence. It has been documented that the fusion peptide domain of influenza HA is a promising candidate target for the prevention/control of viral infection [19,38].

Our previous investigation found that chlorogenin 3-*O*-β-chacotrioside displayed strong inhibitory activity against A/Vietnam/1203/2004(H5N1) and A/Goose/Qinghai/59/05(H5N1) pseudotyped viral entry with IC_50_ of 7.22–9.25 μM, which bore a typical sugar chain of natural spirostan saponins, β-chacotriosyl residue [24]. Structure-effect studies showed the 3-*O*-β-chacotriosyl residue was essential for the activity, and the aglycone structure was as well [24]. To further verify antiviral efficacy against H5N1 viruses, we designed and synthesized a series of novel pentacyclic triterpene derivatives, and utilized the wildtype A/Duck/Guangdong/212/2004 H5N1 virus strain to estimate. The results confirmed that three derivatives XC-27-1, XC-27-2, and UA-Nu-ph-5 exhibited significant dose-dependent inhibitory activities against the wildtype A/Duck/Guangdong/212/2004 (H5N1) virus strain with IC_50_ at 16.83 ± 1.4 μM, 12.45 ± 2.27 μM, and 15.59 ± 2.4 μM, respectively. Interestingly, antiviral activity of XC-27-1 and XC-27-2 against A/Thailand/kan353/2004(H5N1) pseudotyped viruses also presented a significant dose-dependent inhibitory with IC_50_ at 12.76 ± 7.01 μM, 18.47 ± 1.82 μM, respectively. In addition, the plaque reduction neutralization test (PRNT) also manifested that XC-27-1 and XC-27-2 can protect MDCK cells from infection damage. Anyway, the results of these activity assays kept consistent, which suggested chlorogenin 3-O-β-chacotrioside derivatives were potential broad-spectrum H5N1 virus inhibitors.

However, compared with NAIs, the inhibitory activities of these derivatives against H5N1 were relatively lower. Osetamivir and peramivir had the IC_50_ values against H5N1 viral NA with 8.87 and 0.668 nM, according to our study results. However, an NA-R292K mutation that confers broad-spectrum resistance to NA inhibitors has been documented in H5N1 patients after treatment [39,40]. H5N1 viruses with NA-H275Y mutation were recognized to confer extremely high resistance to oseltamivir [41]. No matter how, these derivatives targeting HA may generate the synergy effect against H5N1 viruses in combination with NA inhibitors, particularly to NAI-resistant H5N1 virus.

In order to further confirm the antiviral target of these derivatives aimed at HA, we surveyed HA inhibitory rates of the derivatives against H5N1 pseudo-typed viruses. The study data showed that the inhibitory percentages of the derivatives against H5N1 pseudo-typed viruses were much stronger than that against NA at the same concentration. Therefore, it is evident that the derivatives mainly act on the envelope protein HA of H5N1 viruses. Further studies through PRNT indicated that three derivatives had more strongly reduced the viral plaques forming amounts before virus entry rather than after virus entry, moreover, the inhibitory efficacy presented a dose-response manner, prompting that these derivatives interfere with early stage of H5N1 viral infection, providing protection for MDCK cells from HPAI H5N1 virus infection. This also was supported by the binding affinity data from BLI. The study results confirmed that these derivatives can selectively target at HA of H5N1 virus, preventing viral entry.

To elucidate the interaction of the derivatives and HA, we applied homology modeling and molecular dynamics simulations to predict potential binding patterns and docking sites. Furthermore, on the basis of computational modeling analysis, we verified these predicted docking sites including I48, L329, T331, W367, T387, I391, V394, and T395 with the site-directed mutation methods. Interestingly, the IC_50_ values of the derivatives against mutant A/Thailand/Kan 353/2004 H5N1 pseudo-typed viruses were increased by more than five-folds when the amino acid residue I391 and T395 of HA were substituted. The results implied that these sites were most likely generated interaction between HA and the derivatives. To our surprise, we found that the validated binding sites were very similar to the position of the monoclonal antibody CT149 binding with HA (Appendix A) [42]. It was reported that broad-spectrum antiviral antibody CT149 can recognize the highly conserved epitope region of HA protein which spans two adjacent subunits according to the confirmed crystal structure of the CT149-HA complex [42]. However, the result of hemagglutinin inhibition (HI) assay suggested that the derivatives did not generate interaction with the sialic acid receptor binding site (RBD) of H5N1 viruses. Put together, it can be inferred that the docking sites within HA for chlorogenin 3-*O*-β-chacotrioside derivatives possibly lay in highly conservative regions, and inhibited H5N1 viral entry by targeting HA2 fusion machinery.

In conclusion, the present study demonstrated that chlorogenin 3-*O*-β-chacotrioside derivatives had a strong antiviral activity against HPAI H5N1 viruses, and provided protection for susceptible cells from infection. The mechanism studies suggested that the derivatives can effectively interfere with viral entry at the early stages of H5N1 virus infection, and it is possibly due to generating specific binding to the highly conservative regions within HA. These findings identified that the derivatives were effective H5N1 virus entry inhibitors by targeting HA2 fusion machinery. Despite these encouraging results, further investigation is warranted to elucidate targeting at HA2 inhibitory machinery due to the limitations of the methods used in this study to some extent.

To our knowledge, chlorogenin3-*O*-β-chacotrioside derivatives belong to small molecule pentacyclic triterpene derivatives. Actually, it had been previously reported that pentacyclic triterpenoids Y3 inhibited A/WSN/33 strain (H1N1) with EC_50_ values 2.74–7.41 µM via target viral envelope HA [43]. In addition, it was reported that pentacyclic triterpene derivative (Saikosaponin A) attenuated the replication of H5N1, H1N1, and H7N9 strains through both down-regulation of NF-κB signaling and caspase 3-dependent virus ribonucleoprotein nuclear export [32]. No matter how, these derivatives possibly can produce antiviral effects on other subtype influenza A viruses and NAI-resistant virus strains, more comprehensive scientific assessments would be taken into consideration for further study in the near future.

## Figures and Tables

**Figure 1 viruses-12-00304-f001:**
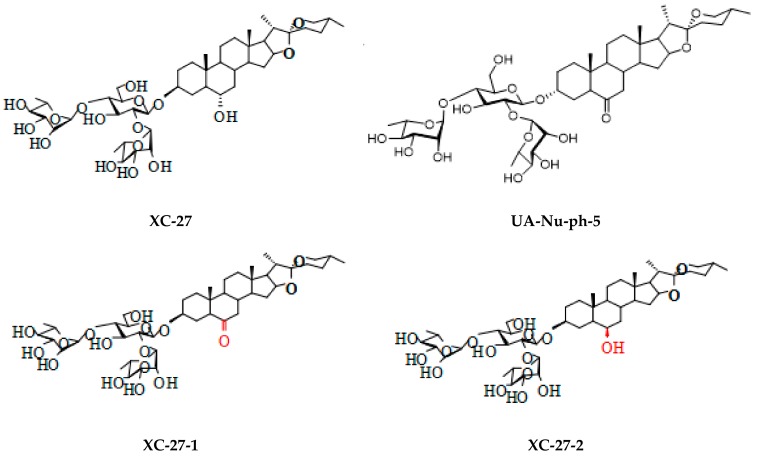
Chemical structures of chlorogenin 3-*O*-β-chacotrioside derivatives.

**Figure 2 viruses-12-00304-f002:**
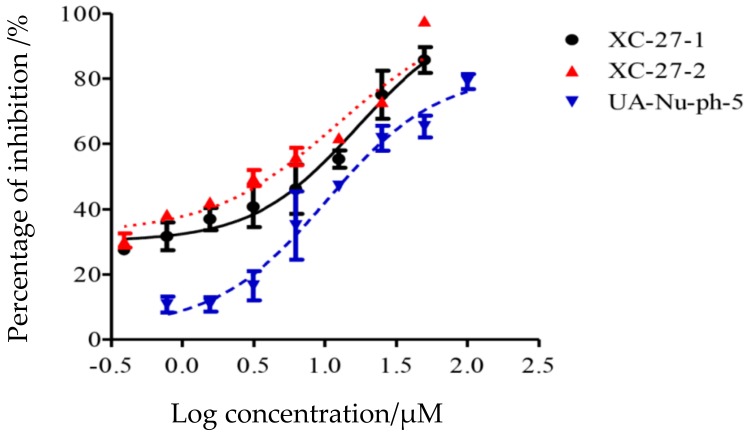
Dose–response inhibition curves of the derivatives against H5N1 avian influenza virus. Three curves represent the dose-response curve of XC-27-2, XC-27-1, and UA-Nu-ph-5 against H5N1 virus from left to right in turn. The error bars represent mean ± standard deviation (three replicates). The horizontal axis represents the log concentration of the derivatives, and the vertical axis represents the inhibition percentage.

**Figure 3 viruses-12-00304-f003:**
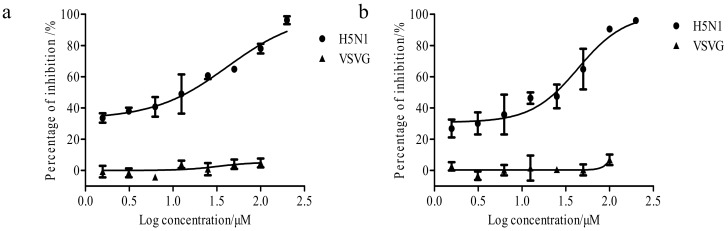
Dose–response inhibition curves of the derivatives against H5N1 and VSVG pseudoviruses. (**a**,**b**) represent the dose-response curves of XC-27-1, XC-27-2 against pseudoviruses, respectively. The error bars represent mean ± standard deviation (three replicates). The horizontal axis represents the log concentration of the derivative, and the vertical axis represents the inhibition percentage.

**Figure 4 viruses-12-00304-f004:**
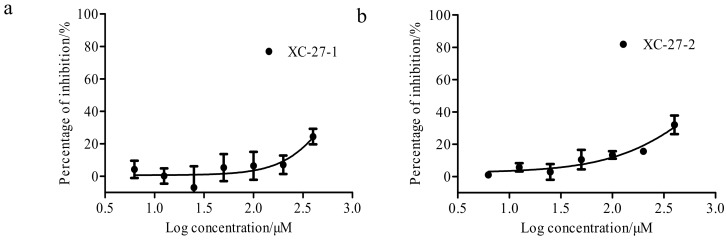
Dose–response curves of the derivatives inhibiting the neuraminidase activity of H5N1 pseudovirus. (**a**,**b**) represent the dose-response inhibition curves of XC-27-1, XC-27-2 to NA, respectively. The error bars represent mean ± standard deviation (three replicates). The horizontal axis represents the log concentration of the derivative, and the vertical axis represents the inhibition percentage.

**Figure 5 viruses-12-00304-f005:**
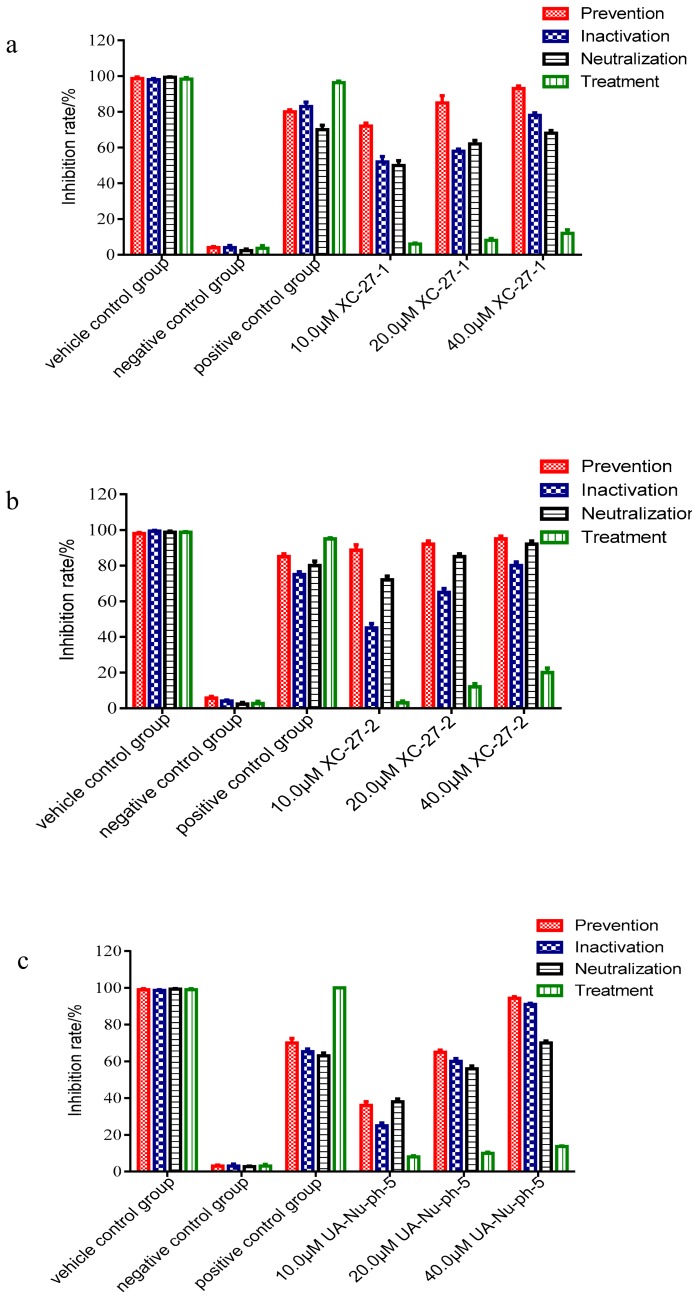
The inhibition rate of derivatives for the H5N1 viral plaques forming amounts. (**a**–**c**) respectively represent the average inhibition rate of XC-27-1, XC-27-2, and UA-Nu-ph-5 for H5N1 viral plaque-forming units (PFUs). Each group was investigated in triplicate for each experiment. Data pooled from three independent experiments (*n* = 3, mean ± S.E.M.).

**Figure 6 viruses-12-00304-f006:**
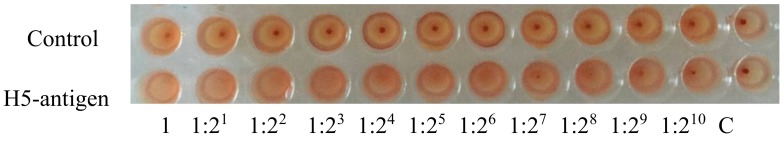
Assay of hemagglutinin titer for the H5N1 virus antigen. For the antigen group, the H5 antigen was added to the wells from the 1st to the 11th column. Among them, the 1st well had the original solution added, and the wells from the 2nd to the 11th column were two-fold serial diluted in turn, the final well had PBS added.

**Figure 7 viruses-12-00304-f007:**
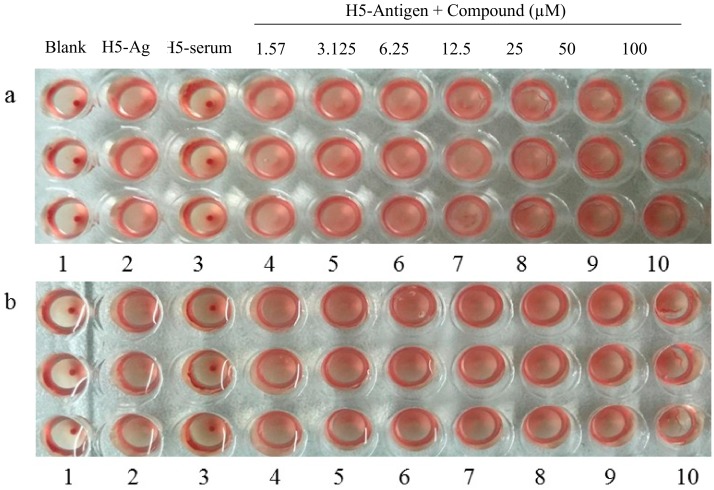
The hemagglutination inhibition effects of derivatives. (**a**,**b**) respectively represent the activity of XC-27-1 and XC-27-2 in inhibition of H5N1 influenza virus-induced aggregation of chicken erythrocytes. In the plates, the H5-antigen group in the the 2nd column had the H5 antigen added. The positive control group in the 3rd column had the H5 serum added. From the 4th to the 10th column, wells were treated from left to right with XC-27-1 and XC-27-2 at 1.57, 3.125, 6.25, 12.5, 25, 50, 100 μM, respectively. The results of the hemagglutination inhibition assay are shown in the photograph of Figure 7. The experiments were repeated independently at least three times.

**Figure 8 viruses-12-00304-f008:**
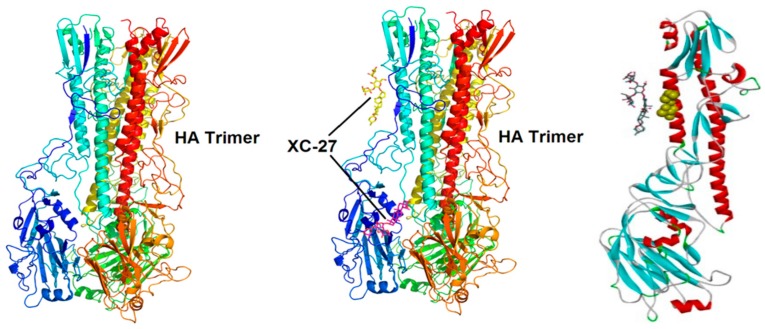
Computational structural prediction for potential binding sites between the derivative XC-27 and HA protein. Using the hemagglutinin crystal structure (2IBX) of the virus VN1194 (H5N1) as a reference structure, all of the available HA sequences and highly homologous HA crystal structure were retrieved from NCBI and PDB databases respectively, based on this, accurate 3D space structures of HA were constructed by using Modeller 9.x software, the molecular docking between XC-27 and HA was performed using AutoDock software.

**Figure 9 viruses-12-00304-f009:**
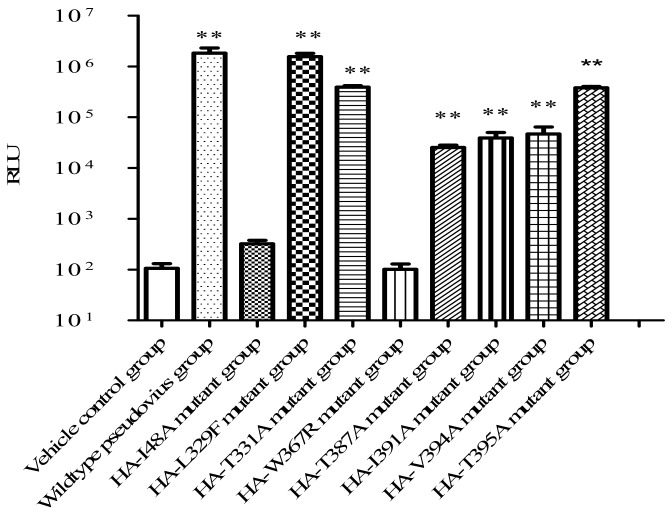
The infectivity of mutant H5N1 pseudovirus strains to MDCK cells. The infection ability of HA mutant pseudoviruses to MDCK cells after co-transfected for 48 h was measured using the luciferase assay method. The RLU of HA mutant pseudovirus at 10-fold diluted are means of three independent experiments (*n* = 3, mean ± S.E.M.). Difference was considered statistically significant when ** *p* < 0.01 vs. vehicle control group.

**Figure 10 viruses-12-00304-f010:**
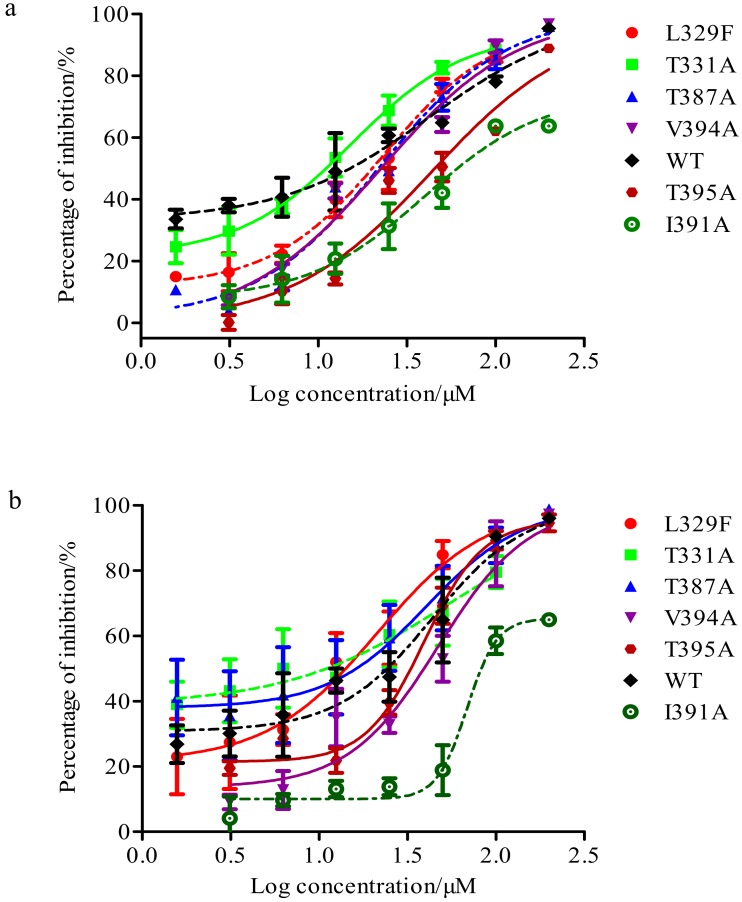
Inhibition rate curves of compounds against H5N1 pseudoviruses. (**a**,**b**) represent the dose–response curves of XC-27-1, XC-27-2 against wildtype and mutant H5N1 pseudovirus strains, respectively. The error bars represent mean ± standard deviation (three replicates). The horizontal axis represents the log concentration of the derivatives, and the vertical axis represents the inhibition percentage.

**Table 1 viruses-12-00304-t001:** Antiviral activities of the tested compounds against H5N1 viruses in MDCK cells ($\bar{x}$ ± S, n = 3).

Compounds	CC_50_ (µM)	IC_50_ (µM)	SI
ribavirin	132 ± 0.87	37.20 ± 1.6	3.55
UA-Nu-ph-5	127 ± 1.697	15.59 ± 2.4	8.15
XC-27-1	1247 ± 1.16	16.83 ± 1.45	74.1
XC-27-2	73.78 ± 0.79	12.45 ± 2.27	6
GA-NEG_2_-5	93 ± 1.45	>100	NA
WEW-35	76 ± 0.79	>100	NA
UA-Me-5	3.125 ± 0.65	3.914 ± 0.214	0.8
i-OA-Me-18	106 ± 0.45	>100	NA
S-100	68 ± 0.729	>100	NA
S-120	57 ± 1.32	>100	NA
S-130	100 ± 1.817	>100	NA

Notes: NA represents no activity.

**Table 2 viruses-12-00304-t002:** The binding affinity parameters for the interaction between the derivatives and HA.

Compounds	Concentration (µM)	Response Value/nm	Association Rate Constant (K_a_) L·(mol·s)^−1^	Dissociation Rate Constant (K_d_) S^−1^	Affinity (K_D_) mol·L^−1^
XC-27-1	3.91	0.0027	1.22 × 10^−1^ ± 0.002	4.76 × 10^2^ ± 0.26	2.57 × 10^−4^ M
	15.6	0.0080			
	31.3	0.0152			
	62.5	0.0376			
	125	0.0501			
XC-27-2	15.6	0.0064	3.33 × 10^−1^ ± 0.001	9.07 × 10^2^ ± 0.17	3.67 × 10^−4^ M
	31.3	0.0104			
	62.5	0.0254			
	125	0.0411			

The binding affinity parameters for the interaction at the gradient concentration of the derivatives were shown in Table 2. The rate constant values of association and dissociation for interaction are means of three independent experiments (*n* = 3, mean ± S.E.M.).

**Table 3 viruses-12-00304-t003:** The infectivity of mutant H5N1 pseudo-viruses with 10-fold diluted to MDCK cells ($\bar{x}$ ± S, n = 3).

Groups	HA Substitution	Relative Light Unit (RLU)
Vehicle control group	—	106 ± 26
wild-type H5N1pseudovirus group	—	1839830 ± 463773 **
HA mutant pseudovirus strains	I48A	323 ± 56
	L329F	1567612 ± 245617 **
	T331A	392427 ± 27535 **
	W367R	102 ± 28
	T387A	25407 ± 2906 **
	I391A	38872 ± 10981 **
	V394A	46849 ± 17278 **
	T395A	379975 ± 26134 **

The infectivity ability was measured by the luciferase assay method. The RLUs of pseudoviruses 10-fold diluted are means of three independent experiments (*n* = 3, mean ± S.E.M.). Difference was considered statistically significant when ** *p* < 0.01 vs. vehicle control group.

**Table 4 viruses-12-00304-t004:** The inhibitory rates of XC-27-1 against mutant H5N1 pseudotyped viruses (%) ($\bar{x}$ ± S, n = 3).

Compound Concentration (µM)	Pseudotyped Virus Groups						
	Mutant Strains with Residues Substituted within HA						Wildtype Strain
	HA-L329F	HA-T331A	HA-T387A	HA-I391A	HA-V394A	HA-T395A	
1.56	14.99 ± 1.00	24.67 ± 5.37	—	—	—	—	33.59 ± 5.25
3.13	16.28 ± 6.13	29.66 ± 13.16	3.80 ± 2.80	8.46 ± 6.44	7.51 ± 3.19	0.16 ± 4.17	38 ± 3.24
6.25	22.22 ± 4.83	37.67 ± 4.11	12.54 ± 3.55	14.17 ± 13.10	17.21 ± 3.18	10.15 ± 4.12	40.69 ± 10.88
12.5	39.83 ± 9.7	53.58 ± 10.66	44.00 ± 2.45	20.76 ± 8.53	42.75 ± 4.27	14.37 ± 3.36	48.95 ± 11.65
25	53.24 ± 7.56	68.71 ± 8.45	49.29 ± 2.34	31.28 ± 12.14	48.09 ± 0.98	46.11 ± 6.96	60.68 ± 3.73
50	76.88 ± 3.78	82.67 ± 3.27	73.02 ± 7.34	42.10 ± 8.41	64.22 ± 4.16	50.47 ± 7.96	64.81 ± 2.96
100	86.23 ± 3.08	88.59 ± 2.30	85.18 ± 5.23	63.79 ± 2.07	89.50 ± 3.39	62.02 ± 1.79	78.01 ± 2.16
200	—	—	96.04 ± 0.02	63.68 ± 0.48	96.84 ± 1.42	88.71 ± 0.68	95.32 ± 3.6

The inhibitory rates (%) of XC-27-1 against pseudovirus strains after treated for 48 h with two-fold serial diluted solution. Data pooled from three independent replicates showed the means values of inhibitory rates of XC-27-1. Each group had 6 parallel wells.

**Table 5 viruses-12-00304-t005:** The inhibitory rates of XC-27-2 against mutant H5N1 pseudotyped viruses (%) ($\bar{x}$ ± S, n = 3).

Compound Concentration (µM)	Pseudotyped Virus Groups						
	Mutant Strains with Residues Substituted within HA						Wildtype Strain
	HA-L329F	HA-T331A	HA-T387A	HA-I391A	HA-V394A	HA-T395A	
1.56	23.02 ± 9.94	38.92 ± 2.22	—	—	—	—	26.83 ± 9.96
3.13	27.42 ± 4.76	43.18 ± 6.69	35.52 ± 13.61	4.11 ± 11.63	21.46 ± 2.79	19.52 ± 3.60	30.10 ± 12.17
6.25	31.31 ± 8.18	50.05 ± 10.83	41.89 ± 15.35	9.72 ± 3.25	20.56 ± 8.63	28.59 ± 0.63	35.80 ± 12.13
12.5	52.03 ± 5.24	47.39 ± 9.67	47.32 ± 9.69	13.11 ± 4.25	38.34 ± 14.10	21.80 ± 6.48	46.30 ± 6.37
25	59.30 ± 4.12	60.42 ± 7.46	59.39 ± 7.44	13.79 ± 4.44	33.91 ± 3.92	39.63 ± 6.48	47.43 ± 13.16
50	84.86 ± 7.29	67.24 ± 7.73	71.59 ± 7.10	18.88 ± 13.26	55.61 ± 11.31	69.24 ± 9.54	64.88 ± 12.50
100	92.22 ± 2.54	79.54 ± 8.49	87.71 ± 9.4	58.51 ± 7.04	94.92 ± 3.35	89.28 ± 4.48	90.53 ± 2.35
200	—	—	98.83 ± 0.02	64.99 ± 1.00	97.32 ± 1.41	94.61 ± 2.59	95.98 ± 0.68

The inhibitory rates (%) of XC-27-2 against pseudovirus strains after treated for 48 h with two-fold serial diluted solution. Data pooled from three independent replicates showed the means values of inhibitory rates of XC-27-2. Each group had 6 parallel wells.

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
