# Peer review of "Study on the Antiviral Activities and Hemagglutinin-Based Molecular Mechanism of Novel Chlorogenin 3-O-β-Chacotrioside Derivatives against H5N1 Subtype Viruses"

_viruses, 2020, doi:10.3390/v12030304_

Round 1
Reviewer 1 Report
Although the manuscript has been improved, there are still many grammar mistakes. The authors should proofread the manuscript.
The following sentence should be still be modulated "Furthermore, H5N1 virus strains are extremely infectious and can be transmitted from avian species to human through multiple pathways."
First, the authors should say the transmission from avian to human is low, although may be potencially possible.
Table 4 can be deleted and explain in the text.
Author Response
Response to the Editor’s and Reviewers’ comments
Dear editors and reviewers:
Thank you for your letter and for the reviewers’ comments concerning our manuscript entitled "Study on the antiviral activities and hemagglutinin-based molecular mechanism of novel chlorogenin 3-O-β-chacotrioside derivatives against H5N1 subtype viruses" (ID: viruses-726367). Those comments are all valuable and very helpful for improving our paper, as well as the important guiding significance to our researches. We again carefully read the comments and made revisions which we hope to meet with approval. Revised portion were marked in red in the paper.The main corrections in the paper and the responds to the reviewer’s comments are as following.
Reviewer #1:
Review Comments and suggestions:
- Although the manuscript has been improved, there are still many grammar mistakes. The authors should proofread the manuscript.
Response:
We sincerely apologize for our grammar errors in our manuscript. We again carefully checked and corrected it in our revised manuscript.
- The following sentence should be still be modulated "Furthermore, H5N1 virus strains are extremely infectious and can be transmitted from avian species to human through multiple pathways."First, the authors should say the transmission from avian to human is low, although may be potencially possible.
Response:
We really appreciate that the reviewer reviewed our manuscript with rigor. What the reviewer said was right. We had corrected this description in the revised manuscript.The descriptionwas expressed as follows.“Furthermore, H5N1 virus strains are extremely infectious and can be transmitted from avian species to human through multiple pathways, although transmission of the H5N1 virus from birds to humans is inefficient.”
- Table 4 can be deleted and explain in the text.
Response:
Thanks a lot for the nice suggestion.As suggested, we have deleted table 4 and did explanation in the revised text.
Finally, fully acknowledge all positive points from the reviewers.
Reviewer 2 Report
Answer to the Response no.1
It is not necessary to add other influenza viruses or subtypes if authors meant to confine the antiviral activity to some specific H5N1 strains in the first place. However, authors should clearly mention that the substances have to be further investigated whether they are universal to other subtype or mutant Gs/Gd lineage H5 viruses because the HA protein is highly variable and having frequent mutations.
Although authors made a comparison with existing antiviral drugs in the previous study, the results should be at least mentioned in discussion so that it gives an idea to readers how the novel substances are potent as compared to the drugs targeting NA activity.
Answer to the Response no.2
Authors had better not use the terms, as it said, prevention, inactivation, neutralization, and treatment. First of all, experimental concept of inactivation and neutralization is not DECISIVE because we never know if the derivatives technically whether neutralize or inactivate the viruses by only giving them different contact times. Second, when it comes to treatment, the infected cell should be treated by adding the derivatives in the 1% agarose so that they can take the antiviral effect consistently. Overall, to avoid those misconception, changing those terms with other words, for example x hour pre-contact group and x hour post-infection group, would be more acceptable.
Answer to the Response no.3
The response was not a right answer to the point that why do they have higher effect when the substances have contact with virus before virus entry rather than after virus entry. The fact that they doesn’t show hemagglutinin inhibition function does not account for the difference. The possible explanation/theory would be like that because the derivate alone may not be absorbed into the cells, treatment after virus entry would not have antiviral effect on virus propagation. Again, if the agarose containing the derivatives was overlaid after virus infection, it might have increased inhibition of plaque formation.
Answer to the Response no.4
As authors responded that the methods used in this study have limitations to an extent as to fully embrace the antiviral mechanism of the substances, that comment should be included in discussion for clarification.
Answer to the Response no.5
When we say the reverse genetically generated mutant virus, it refers to a viable virus generated by transfecting single or multiple plasmids expressing all eight genes of viruses into a cell such as 293T, not the pseudo-virus. Please answer this question properly.
Answer to the Response no.6
Thank you for the proper answer.
Answer to the Response no.7
Thank you for the example and kind description.
Answer to the Response no.8
Properly corrected.
Answer to the Response no.9
What factors or steps of the protocol could make differences in the CC50 value of RBV to MDCK cells? Could you provide more instances that had similar CC50 values to this work? Previous works published by the same setting would be appreciated.
Answer to the Response no.10
Thank you for the kind answer.
Answer to the Response no.11
Thank you for the kind answer.
Answer to the Response no.12
The table is merged properly. In addition, there are more ways to improve readability and make it concise. Overall, all figures should be interpretable in a stand-alone way, pinpointing core materials and methods used in this study. Use the figure caption, try to explain the figure by only reading it.
- Put three plaque reduction assay results in a single figure and give them different indexes such as Figures 9A, 9B, and 9C. Also I see these figures doesn’t have a right title.
- Set aside the primer sets listed in the materials and methods to a supplemental table.
- Put the HI figures together and present the derivatives and concentration treated in the figure by writing next to the actual figure, not by just marking them as the numbers.
- The same as the plaque reduction figure, make those two inhibition rate curves with mutant pseudoviruses in a single figure having subdivisions.
Author Response
Response to the Editor’s and Reviewers’ comments
See the attachment

This manuscript is a resubmission of an earlier submission. The following is a list of the peer review reports and author responses from that submission.
Round 1
Reviewer 1 Report
Viruses-681515
Wan-zhen and coworkers assessed the inhibitory effect of cholorogenin 3-O-b-chacotrioside derivatives against HPAI H5N1 viruses and they tried to elucidate the potential molecular mechanism underlying the antiviral action. They found 3 derivatives (UA-Nu-ph-5, XC-27-1 and XC-27-20) that strongly inhibited wild-type A/Duck/Guangdong/212/2004 H5N1 virus and they have demonstrated that these compounds inhibit the virus by blocking the hemagglutinin function.
The manuscript needs important proofreading as it contains many fundamental grammar errors and punctuation mistakes. In addition, the techniques are written with basic details that the reviewer does not consider necessary for the readers. The experimental design is acceptable but Materials and Methods should be completely re-written. Result section should be simplified and some table/figures should be eliminated.
General comments:
Introduction: Line 54-55: What do you mean with “can transmit from avian to human through multiple channels”? Also, in general avian influenza viruses have a low transmissible efficacy to humans. Why do you compare it with SARS? Line 80-81: What are “broad anti-influenza H5N1 viruses”
Material and methods:
This section should be re-written in a shorter way. The reviewer will only provide some examples of things to change/delete.
Cell lines and chemical reagents section should be shortened. No need of such a detail. Virus amplification should be deleted. Just mention that viruses were amplified in eggs. Line 176: Why 10 ul of MTT was used instead of 22 as in line 133? Line 217: No need to explain how to prepare a 1% chicken RBCs nor the whole technique. Please simplify. Line 249: What are the “international SOPs”? Line 251-252: If stock concentration was 400 uM, and you did 2-fold dilution, did you really have 200 uM, 100 uM etc.? You need to consider all reagents added to the well. NA enzyme and compounds concentration should be expressed in concentration and not in ul. Line 280: Why did you change to A/Thailnad/kan353/2004? Plasmid transfection and amplification; Plasmid extraction and purification; Gene sequencing; Site-directed mutagenesis should be deleted. Statistical analysis is confusing; did you use ANOVA follow by t-test? Why? Line 473: “should be combined” Did you mean “were combined”?.
Results
Table 2 should be deleted. Figure 3, 5 and 7 are not necessary as figures 4, 6 and 8 are clear. The different treatments (prevention, inactivation, neutralization and treatment) should be defined. Same with figure 9 and table 5, keep one way to represent data. The authors should consider simplify the data as much as possible as there are too many tables and figures. Some data could be shown as supplementary data.
Discussion:
The discussion section is very poor and needs to be re-written. With the amount of data presented, the authors should discuss further their findings. Lines 868-910 should be placed in the introduction section.
Author Response
Response to the Editor’s and Reviewers’ comments
Dear editors and reviewers:
Thank you for your letter and for the reviewers’ comments concerning our manuscript entitled "Study on the antiviral activitiesand hemagglutinin-based molecular mechanism of novel chlorogenin 3-O-β-chacotriosidederivativesagainst H5N1 subtype viruses" (ID: viruses-681515). Those comments are all valuable and very helpful for improving our paper, as well as the important guiding significance to our researches. We had carefully read the comments and made revisions which we hope to meet with approval. Revised portion were marked in red in the paper. The main corrections in the paper and the responds to the reviewer’s comments are as following.
Reviewer #1:
Review Comments and suggestions:
1.The manuscript needs important proofreading as it contains many fundamental grammar errors and punctuation mistakes. In addition, the techniques are written with basic details that the reviewer does not consider necessary for the readers.The experimental design is acceptable but Materials and Methods should be completely re-written.Result section should be simplified and some table/figures should be eliminated.
Response:
We really appreciate that the expert reviewed our manuscript with carefulness and responsibility. We sincerely apologize for our grammar errors and punctuation mistakes in our manuscript. Thanks a lot for the nice suggestion.We corrected it in our revised manuscript.
2.Introduction: Line 54-55: What do you mean with “can transmit from avian to human through multiple channels”?
Response:
Thank you so much for reminding us. We sincerely apologize for the confusing caused to you. We had corrected the mistakes in the revised manuscript. The word“multiple channels” was changed to “multiple pathways”.Here, we mean that human possibly infect HPAI H5N1 virusesfrom avian throughhuman themselves digestive and respiratory tracts,eye conjunctiva and broken skinetc.
3.Also, in general avian influenza viruses have a low transmissible efficacy to humans. Why do you compare it with SARS?
Response:
Yes, we totally agree with this point.We had deleted this sentence “far more beyond than that of SARS”. In fact, we intend to emphasize the harmfulness of H5N1 infectionby comparison, because people felt deeply about the worldwide outbreak of SARS epidemic.
4.Line 80-81: What are “broad anti-influenza H5N1 viruses”?
Response:
Thank you for the kind reminds. We sincerely apologize for the confusing caused to you.We had corrected the mistakes on “broad anti-influenza H5N1 viruses”in the revised manuscript.The descriptionwas expressed as follows: “antiviral against multiple H5N1 subtype strains”.
This section should be re-written in a shorter way. The reviewer will only provide some examples of things to change/delete.Cell lines and chemical reagents section should be shortened. No need of such a detail.Virus amplification should be deleted. Just mention that viruses were amplified in eggs.
Response:
Thanks for the good suggestion. We did carry out modifications according to the suggested.
6.Line 176: Why 10μl of MTT was used instead of 22μl as in line 133?
Response:
Thanks a lot. The reviewer read so carefully about our manuscript and thanks for your time!We sincerely apologize for the confusing caused to you. Generally, in MTT assay, 5mg/mL MTT was added to medium at the final concentration of 0.1%each well of96-well plate. Due to the change of final volume, so 10μl of MTT was used to 0.1mL mediuminstead of 22μl.
7.Line 217: No need to explain how to prepare a 1% chicken RBCs nor the whole technique. Please simplify.
Response:
Thanks a lot for the nice suggestion.We simplified the methods of “the preparation of 1%(v/v) chicken RBCs suspension”in the revised manuscriptaccording to the suggested.
8.Line 249: What are the “international SOPs”?
Response:
Thank you for the kind reminds. We sincerely apologize for the confusing caused to you. We had corrected the mistakes on “international SOPs”in the revised manuscript.The descriptionwas expressed as follows: “National Standard Operating Procedures (SOP).For example,
9.Line 251-252: If stock concentration was 400 μM, and you did 2-fold dilution, did you really have 200 μM, 100 μM etc.? You need to consider all reagents added to the well.
Response:
The reviewer read so carefully about our manuscript and thanks for your time! We sincerely apologize for the error of English writing.The sentence"The compounds were dissolved in DMSO to gain a 400µM concentration solution, and then the solution was two-fold serial diluted with 2× assay buffer to a final concentration of 200µM, 100µM, 50µM, 25µM, 12.5µM to 6.25µM" was changed to "The compounds were dissolved in DMSO to gain a 4mM concentration solution, and then the solution was two-fold serial diluted with ddH2O to a final concentration of 2mM, 1mM, 0.5mM, 0.25mM, 0.125mM to 0.0625mM” in the revised manuscript.The reaction system is as follows.
|
System |
Well number |
||||||
|
1 |
2 |
3 |
4 |
5 |
6 |
7 |
|
|
NA Buffer |
70µl |
70µl |
70µl |
70µl |
70µl |
70µl |
70µl |
|
NA |
10µl |
10µl |
10µl |
10µl |
10µl |
10µl |
10µl |
|
Compoundvolume |
4mM, 10µl |
2mM, 10µl |
1mM, 10µl |
0.5mM, 10µl |
0.25mM,10µl |
0.125mM, 10µl |
0.0625mM, 10µl |
|
ddH2O |
0µl |
0µl |
0µl |
0µl |
0µl |
0µl |
0µl |
|
Fluorescence substrate |
10µl |
10µl |
10µl |
10µl |
10µl |
10µl |
10µl |
|
Total volume |
100µl |
100µl |
100µl |
100µl |
100µl |
100µl |
100µl |
|
Compound concentration |
400µM |
200µM |
100µM |
50µM |
25µM |
12.5µM |
6.25µM |
10.NA enzyme and compounds concentration should be expressed in concentration and not in μl.
Response:
Yes, we totally agree with this point. The concentration of NA enzyme and compounds should be expressed by μM or M. In fact, we performed the assay of neuraminidase inhibitoryactivitiesaccording to the manufacturer’sinstructions. In this manuscript, we described in detail the operation procedure. But due to Kit formulation intellectual property, the manufacturedidn’t provide the specific concentration about NA enzyme. We sincerely apologize for it.
11.Line 280: Why did you change to A/Thailnad/kan353/2004?
Response:
That’s really a nice question.Actually, in this study, we appliedA/Duck/Guangdong/212/2004
(H5N1) isolated strainto perform plaque reduction neutralization test and IC50 value determination. However, consideringthe genes stabilityand strong infectivity to host cells,we changed to selectA/Thailand/kan353/2004(H5N1) to perform the analysis of site-directed mutagenesiswith a pseudovirion system.
12.In “material and methods” section,Plasmid transfection and amplification; Plasmid extraction and purification; Gene sequencing; Site-directed mutagenesis should be deleted. Response:
Thanks for the good suggestion. We did carry out some deletion on “Plasmid transfection and amplification; Plasmid extraction and purification; Gene sequencing; Site-directed mutagenesis”according to the suggested.We hopethat the modificationsto meet with approval.
13.Statistical analysis is confusing. Did you use ANOVA follow by t-test? Why? Line 473: “should be combined” Did you mean “were combined”?
Response:
Thank you for the kind reminds. We sincerely apologize for the error of English writingand that confusing caused to you.The sentence"The statistical results are presented as the mean ± standard deviation (SD). The comparison of the measured variables between groups was performed using One-way ANOVA method combined with a t-test and so on" was changed to "The statistical results are presented as the mean ± standard deviation (SD). The comparisonbetween groups wasconducted using analysis of variance (ANOVA)"in the revised manuscript.
14.Table 2 should be deleted.Figure 3, 5 and 7 are not necessary as figures 4, 6 and 8 are clear.
Response:
That’s really a great suggestion. As suggested, we have deleted table 2 in the revised manuscript. The results were very clearly showed in table 4, 6 and 8 indeed, we quite agreed with opinion thatit was not necessary for figure 3, 5 and 7. Considering that figure 3, 5 and 7 are more intuitive and authentic, and it might make the readers more understand once it published.We have put figure 3, 5 and 7 into the supplementary data.
15.The different treatments (prevention, inactivation, neutralization and treatment) should be defined.
Response:
That’s really a great suggestion.We appreciatedso much that the reviewer pointed it out, otherwise it mightconfuse the readers once it published.A shortdescription for the definition of four infection manners was added in the revised manuscript. The descriptionwas expressed as follows.
“Briefly,cells are treated with derivativesfor 2 h before adding viruses, which was ‘prevention’. Viruses are treated with derivativesfor 30min before adding to normal cells, which was considered as ‘inactivation’. Viruses and derivatives are added to cells at the same time and incubated for 1 h, which was defined as‘neutralization’. Cells were infected with viruses for 1h before adding derivatives, which was considered 'treatment'.”
Reference:
[1] ZHAO Lihua,CHEN Quanjiao .Anti-viral Effects of Qingkailing Injection Against Influenza H1 N1 ,H5N1 ,and H7N9 Virus in Vitro(Article in China). Traditional Chinese Drug Research & Clinical Pharmacology. 2015, 26 (5):644-648
[2] Yingsakmongkon S, Miyamoto D, Sriwilaijaroen N, FujitaK, Matsumoto K, Jampangern W , et al.In vitro inhibition of human influenza A virus infection by fruit-juice concentrate of Japanese plum (Prunusmume SIEB. et ZUCC).Biol Pharm Bull. 2008, 31(3):511-5.DOI: 10.1248/bpb.31.511
[3]Miyamoto D, Hasegawa S, Sriwilaijaroen N, Yingsakmongkon S, Hiramatsu H, Takahashi T, et al.Clarithromycin inhibits progeny virus production from human influenza virus-infected host cells.Biol Pharm Bull. 2008, 31(2):217-22.DOI: 10.1248/bpb.31.217
16.Same with figure 9 and table 5, keep one way to represent data.The authors should consider simplify the data as much as possible as there are too many tables and figures. Some data could be shown as supplementary data.
Response:
We really appreciate the goodsuggestion. As to suggested, we did carry on the deletion of figure 9.
17.The discussion section is very poor and needs to be re-written. With the amount of data presented, the authors should discuss further their findings. Lines 868-910 should be placed in the introduction section.
Response:
Thank you for the kind reminds. We sincerely apologize for the poor discussion section.We had carefully made revisions. As suggested, lines 868-910 were edited into the introduction section.
Finally, fully acknowledge all positive points from the reviewers.
Reviewer 2 Report
In the present manuscript entitled “Study on the antiviral activities and hemagglutinin-based molecular mechanism of novel chlorogenin 3-O-β-chacotrioside derivatives against H5N1 subtype viruses”, Shi et al. suggested that UA-Nu-ph-5, XC-27-1 and XC-27-2 as chemical derivatives of XC-27 inhibit infection of avian influenza virus H5N1 by directly targeting viral HA2 fusion protein. Their mechanism as antiviral reagents is interesting. However, scientific rationale for mode-of-antiviral action is not solid and manuscript, including data presentation, is not organized as mentioned below.
Major comments:
Antiviral activity of the three main compounds should be evaluated against different (sub)types of influenza A and B viruses as well as additional H5N1 isolates in the cell-based assay system. In all EC50 determination experiments, at least 2 or 3 well-evaluated antiviral compounds, such as oseltamivir carboxylate, ribavirin (RBV) or amantadine, should be included to show experimental reliability. In Figs 3 to 8, RBV that has been known to inhibit viral polymerase activity was used as a positive control. However, it targeted all steps, prevention, inactivation, neutralization and treatment. It should be checked whether this time-of-addition experiment was optimized. What is the difference between inactivation and neutralization. Experimental scheme should be described in more detail in Materials and Methods. The authors suggested that the active compounds target HA2. Why do they inhibit the prevention step rather than the inactivation or neutralization step? As a direct evidence, HA2-mediated fusion assay showing syncytia formation should be compared in the presence of the compounds by using a control reagent such as anti-HA2 antibody or arbidol or other known HA2 inhibitors. Pseudovirus internalization assay using mutants suggested that interaction of XC-27-1 or XC-27-2 to the 391th amino acid, isoleucine, within HA is critical. Why not test reduction in their antiviral activity against reverse genetically generated mutant virus with HA-I391A substitution or decrease in their binding to the mutant HA protein through the binding affinity analysis?
Minor comments:
Abstract: There are so many abbreviations, such as HI, NAI or BLS, which are not explained. SI above 2 is not an enough value to define antiviral efficacy. Line 54: It has a grammatical error, “an extremely infectious and…”. Fig. 1: Chemical structure of the compounds is ambiguous and unclear. Table 1: Generally, CC50 value of RBV to MDCK cells is higher than the presented one. Line 163: Why did the authors observe the CPE with a microscope, not by cell viability measurement using MTT? Materials and Methods: It is too lengthy and too detailed. Reference citations seem to be enough in most of the sections. Tables 1 and 3: The numbers should be summarized in same significant digits. Table 2: Quantitative measurement is required. Figures 3, 5, 7: Plaque titrations should be performed again to make them countable. Tables 6 and 7, and Figure 10: It is recommended to combine Tables 6 and 7. No data with 200 micromolar concentration of the compounds against VSVG. Fig 13, Figure legend: Fig11-b to Fig13-b
Author Response
Response to the Editor’s and Reviewers’ comments
Dear editors and reviewers:
Thank you for your letter and for the reviewers’ comments concerning our manuscript entitled "Study on the antiviral activitiesand hemagglutinin-based molecular mechanism of novel chlorogenin 3-O-β-chacotriosidederivativesagainst H5N1 subtype viruses" (ID: viruses-681515). Those comments are all valuable and very helpful for improving our paper, as well as the important guiding significance to our researches. We had carefully read the comments and made revisions which we hope to meet with approval. Revised portion were marked in red in the paper. The main corrections in the paper and the responds to the reviewer’s comments are as following.
Reviewer #2:
Review Commentsand suggestions:
In the present manuscript entitled “Study on the antiviral activities and hemagglutinin-based molecular mechanism of novel chlorogenin 3-O-β-chacotrioside derivatives against H5N1 subtype viruses”, Shi et al. suggested that UA-Nu-ph-5, XC-27-1 and XC-27-2 as chemical derivatives of XC-27 inhibit infection of avian influenza virus H5N1 by directly targeting viral HA2 fusion protein. Their mechanism as antiviral reagents is interesting. However, scientific rationale for mode-of-antiviral action is not solid and manuscript, including data presentation, is not organized as mentioned below.
1.Antiviral activity of the three main compounds should be evaluated against different (sub)
types of influenza A and B viruses as well as additional H5N1 isolates in the cell-based assay system. In all EC50 determination experiments, at least 2 or 3 well-evaluated antiviral compounds, such as oseltamivir carboxylate, ribavirin (RBV) or amantadine, should be included to show experimental reliability.
Response:
That’s really a nice suggestion.We agreed with this point. Actually, in this study, we did select three well-evaluated antiviral compounds as control, including oseltamivir carboxylate, peramivir and zanamivir, and obtained the antiviral activities with IC50 values 8.87nM, 0.668nM and 1.58nM respectively. Considering that they are NAI agents targeting at neuraminidase, there are different drug target from chlorogenin 3-O-β-chacotrioside derivatives, so these results were not added in this manuscript. The results were as following:
Fig1. Dose–response inhibition curves of these compounds against H5N1viruses
In previous study, we had investigated the antiviral activities of chlorogenin 3-O-β-chacotrioside derivativeXC-27against H5N1viruses includingA/AnHui/1/2005, A/VietNam/1203/2004 and A/Goose/Qinghai/59/05 virus strainswith IC50 values of 8.54 and 6.00 μM[1-3].Moreover, the results determined that the susceptibility of influenzaA/Puerto Rico/8/34(H1N1) virus tochlorogenin 3-O-β-chacotrioside derivativeincreased ten folds than that of H5N1viruses, and had an IC50value of 0.6-1.8μM.
Reference:
[1] Gaopeng Song, Sen Yang, Wei Zhang, Yingli Cao, Peng Wang, Ning Ding, Zaihong, Zhang, Ying Guo, Yingxia Li.Discover of the first series of small molecure H5N1 entry inhibitors. J. Med. Chem. 2009, 52(23):7368-7371
[2] Ding N, Chen Q, Zhang W, Ren S, Guo Y, Li Y.Structure-activity relationships of saponin derivatives: a series of entry inhibitors for highly pathogenic H5N1 influenza virus.Eur J Med Chem. 2012, 53:316-326
[3]Song GP, Shen XT, Li SM, Li YB, Liu YP, Zheng YS, Lin RH, Fan JH, Ye HM, Liu SW. Structure-activity relationships of 3-O-chacotriosyl ursolic acid derivatives as novel H5N1 entry inhibitors. Eur J Med Chem, 2015, 93: 431-442
2. In Figs 3 to 8, RBV that has been known to inhibit viral polymerase activity was used as a positive control. However, it targeted all steps, prevention, inactivation, neutralization and treatment. It should be checked whether this time-of-addition experiment was optimized. What is the difference between inactivation and neutralization. Experimental scheme should be described in more detail in Materials and Methods.
Response:
That’s really a very nice question. Actually,we did have done some literature investigationbefore we performed plaque reduction assay, there have been some reportsonthe four application methods (prevention, inactivation, neutralization and treatment) in thepublished papers.Before formal test, we conductedpreliminaryexperiments. The results determined that the viral absorption occurredwithin 0-2h after adding H5N1 viruses to cells. So we referred to the reportedtime-of-addition. Furthermore, in this study, plaque reduction neutralization test was to further verify that the derivatives effect on HA on basis of the results of H5N1 pseudovirusexperiments. (Yingsakmongkon S, etal. 2008)
In the early stages of viral infection, there are process including virus attachment to host cells and subsequent entry via fusion of viral membrane with a host cell membrane, and fusion of viral membrane with the endosome.Inactivationpattern mainlyinvestigate the ability of the derivativesdirectly inhibiting viruses.Neutralization pattern is to investigate the ability of the derivatives inhibitingviral invasion.
As suggested, a shortdescription for the definition of four infection manners was added in the revised manuscript. The descriptionwas expressed as follows.“Briefly,cells are treated with derivativesfor 2 h before adding viruses, which was ‘prevention’. Viruses are treated with derivativesfor 30min before adding to normal cells, which was considered as ‘inactivation’. Viruses and derivatives are added to cells at the same time and incubated for 1 h, which was defined as‘neutralization’. Cells were infected with viruses for 1h before adding derivatives, which was considered 'treatment'.”
Reference:
[1] ZHAO Lihua,CHEN Quanjiao .Anti-viral Effects of Qingkailing Injection Against Influenza H1 N1 ,H5N1 ,and H7N9 Virus in Vitro(Article in China). Traditional Chinese Drug Research & Clinical Pharmacology. 2015, 26 (5):644-648
[2] Yingsakmongkon S, Miyamoto D, Sriwilaijaroen N, FujitaK, Matsumoto K, Jampangern W , et al.In vitro inhibition of human influenza A virus infection by fruit-juice concentrate of Japanese plum (Prunusmume SIEB. et ZUCC).Biol Pharm Bull. 2008, 31(3):511-5.DOI: 10.1248/bpb.31.511
[3]Miyamoto D, Hasegawa S, Sriwilaijaroen N, Yingsakmongkon S, Hiramatsu H, Takahashi T, et al.Clarithromycin inhibits progeny virus production from human influenza virus-infected host cells.Biol Pharm Bull. 2008, 31(2):217-22.DOI: 10.1248/bpb.31.217
3.The authors suggested that the active compounds target HA2. Why do they inhibit the prevention step rather than the inactivation or neutralization step?
Response:
That’s a very nice question.Actually,our results confirmed that three derivatives XC-27-1, XC-27-2 and UA-Nu-ph-5showed strongest inhibitory effects against HPAI H5N1 virusin the prophylacticadministration pattern.Although the derivatives can also reduce the amounts of viral plaques forming in a neutralization administration patterns, its inhibition effects became weaker than prophylacticadministration pattern.Furthermore, theresult of HI testindicated that the derivatives did not inhibitHA1 proteinwhich contains sialic acid receptor binding site (RBD). So we deduced that the derivatives target at HA2. This does not contradict the result of PRNT.
Hemagglutinin (HA),the major glycoprotein on the surface of influenza virus, is composed of head (HA1) and stem(HA2/HA1) domains.HA is responsible for entry and infection through binding to terminal sialic acids on cellular receptors and mediating fusion.The HA1 head contains a sialic acid receptor binding site, which is responsible for virus binding to the surface of the host cell. So HA1 inhibitor can interfere or block viral attachment. The HA2 stem is a transmembraneprotein, which mediates virus entry by endocytosis into the endosome after viruses binding on the host cell surface. So HA2 inhibitor also actson the early stage of the cells infection.Theirinhibitory effect in the prevention pattern is strongest than others, which coincides with theory.
4.As a direct evidence, HA2-mediated fusion assay showing syncytia formation should be compared in the presence of the compounds by using a control reagent such as anti-HA2 antibody or arbidol or other known HA2 inhibitors.
Response:
The reviewer’ssuggestionwas right. We agreethere should be more direct evident to support derivatives as HA2 inhibitors, such asHA2-mediated fusion assay andcrystal structure study of derivatives in complex with HA,could be taken into consideration tofurther provide“insight”for targeting HA2.However, it is first time that our study confirmed chlorogenin 3-O-β-chacotriosidederivativescould strongly inhibitedwild-type A/Duck/Guangdong/212/2004 H5N1 viruses on the basis of previous pseudovirus system.The focus of this manuscript is the identification of the new generation of compounds, which open up the window for further investigations. Our study could be considered as the very firststep for the mechanism research. There must be some hidden mechanisms associated with HA2 inhibition. We will continue working on the detailed mechanistic studies in the near future since the complicacy might not be fully understood in one study.
5.Pseudovirus internalization assay using mutants suggested that interaction of XC-27-1 or XC-27-2 to the 391th amino acid, isoleucine, within HA is critical. Why not test reduction in their antiviral activity against reverse genetically generated mutant virus with HA-I391A substitution or decrease in their binding to the mutant HA protein through the binding affinity analysis?
Response:
We quite agree with the view of review expert. Actually, the binding affinityof HA-I391A mutant protein and derivatives can be detected through SPR, ITC and BLI etc.But these analysis technologiesonlyare suitable for intermolecular interaction. In present study, we had performed their antiviral activity against reverse genetically generated mutant virus withL329F, T331A, T387A, I391A, V394A and T395A substituted, unexpectedly, we found that the susceptibility of pseudo-virus strain to both derivatives declined when residue I391A substitution. It’s really rule that we may verify it through decrease in their binding to the mutant HA protein. In fact, we are currently working on the construction of HA protein and its mutantby applying the protein expression and purification technology, and also intend to further verify it in the near future.
However, themolecular mechanismmight not be fully understood in one study.We would like to work continuously on the further investigation of the exact mechanism.
6.Abstract: There are so many abbreviations, such as HI, NAI or BLS, which are not explained.
Response:
Thank you so much for reminding us. As suggested, the full names of abbreviationswere added
accordinglyin the abstract section. Revised portion were marked in redfond in the revised
manuscript.
7.SI above 2 is not an enough value to define antiviral efficacy.
Response:
Thank you so much for reminding us. We really appreciate the reviewerfor rigorous attitude.Generally,SI is an index for judging safe range of compound. However, it is also as a marker for antiviral activity in the screening of lead compounds. Actually,in some reported paper, there are some descriptions about SI, according to Al-Salahi R’s reports: “In accordance to the statistical analyses and in terms of SI as a marker for antiviral activity, all tested molecules have been classified into three groups: inactive-(SI<2), active-(2≦SI<10) and very active-types(SI≥10)”(Al-Salahi R,etal. 2016).Whereas, according to Maddry JA’s reports: “The selective index (SI) was calculated as SI = CC50/EC50. The criteria for determining compound activity are based on its SI. Compounds with an SI value of >3 were defined as active, whereas compounds that exhibited an SI value less than 3 were defined as inactive”(Maddry JA, et al. 2011). Based on these literatures, we had corrected this description in the revised manuscript.The description was expressed as following:
“In accordance to the criteria for compound activityinvitro tests,compounds with an SI value of >3 were defined as active. As shown in Table 3, among the synthesized compounds, chlorogenin 3-O-β-chacotrioside derivatives UA-Nu-ph-5, XC-27-1 and XC-27-2 had their SI values more than 6, so it was obvious that these derivatives displayedlow toxicity and better inhibitory activitythan others against HPAI H5N1virus. In particularly, the inhibitory activities of the derivative XC-27-1 showed highly selective.”
Reference:
[1] Al-Salahi R, Abuelizz HA, Ghabbour HA, El-Dib R, Marzouk M. Molecular docking study and antiviral evaluation of 2-thioxo-benzo[g]quinazolin-4(3H)-one derivatives.Chem Cent J. 2016, 10:21. doi: 10.1186/s13065-016-0168-x.
[2]Li Z, Zhan P, Naesens L, Vanderlinden E, Liu A, Du G, De Clercq E, Liu X.synthesis and preliminary biologic evaluation of 5-substituted-2-(4-substituted phenyl)-1,3-benzoxazoles as a novel class of influenza virus A inhibitors.ChemBiol Drug Des. 2012, 79(6):1018-24. doi: 10.1111/j.1747-0285.2012.01344.x.
[3]Maddry JA, Chen X, Jonsson CB, Ananthan S, Hobrath J, Smee DF, Noah JW, Noah D, Xu X, Jia F, Maddox C, Sosa MI, White EL, Severson WE. Discovery of novel benzoquinazolinones and thiazoloimidazoles, inhibitors of influenza H5N1 and H1N1 viruses, from a cell-based high-throughput screen. J Biomol Screen. 2011,16(1):73-81. doi: 10.1177/1087057110384613.
8.Line 54: It has a grammatical error, “an extremely infectious and…”.Fig. 1: Chemical structure of the compounds is ambiguous and unclear.
Response:
We sincerely apologize for our mistakes. We had corrected these mistakesin the revised manuscript.The sentence “Furthermore, H5N1 virus strains have anextremely infectious and can transmit from avian species to human through multiple channels.”was changed to “Furthermore, H5N1 virus strains are extremely infectious and can be transmittedfrom avian species to human through multiple pathways.”We had adjusted the picture in the revised manuscript, and the chemical structure was as following:
Fig.1. Chemical structures of chlorogenin 3-O-β-chacotrioside derivatives
9.Table 1: Generally, CC50 value of RBV to MDCK cells is higher than the presented one.
Response:
What the reviewer said is right.Actually, with regarding to ribavirin, the CC50 value to MDCK cell has been reported in some study papers. According to W. Markland’s reports, it was 232µM(W. Markland, et al. 2000).The CC50 valueof RBV to MDCK cell was >100µM(Li Z,etal.2012).Whereas, Sidwell RW, etal thought that CC50 value of RBV to MDCK cells was 560µg/ml (MW 244.21), that is, it equals 2.293µM (Sidwell RW, et al. 2005). However, the CC50 value of RBV to MDCK cells which was obtained in our study corresponds closely to that of Li Z’sreport.These differences may due to experimental errors.
Reference:
[1]W. Markland, T. J. McQuaid, J. Jain, A. D. Kwong. Broad-Spectrum Antiviral Activity of the IMP Dehydrogenase Inhibitor VX-497: a Comparison with Ribavirin and Demonstration of Antiviral Additivity with Alpha Interferon. Antimicrob Agents Chemother. 2000,44(4): 859-866. doi: 10.1128/aac.44.4.859-866.2000
[2]Sidwell RW, Bailey KW, Wong MH, Barnard DL, Smee DF. In vitro and in vivo influenza virus-inhibitory effects of viramidine. Antiviral Res.2005, 68(1):10-7.
[3] Li Z, Zhan P, Naesens L, Vanderlinden E, Liu A, Du G, De Clercq E, Liu X.synthesis and preliminary biologic evaluation of 5-substituted-2-(4-substituted phenyl)-1,3-benzoxazoles as a novel class of influenza virus A inhibitors.ChemBiol Drug Des. 2012, 79(6):1018-24. doi: 10.1111/j.1747-0285.2012.01344.x.
10.Line 163: Why did the authors observe the CPE with a microscope, not by cell viability measurement using MTT?
Response:
That’s a very nice question.MTT assay and CPE method can be widely applied toevaluate antiviral effects and toxicity of compounds in vitro. Moreover, MTT assay combined with CPE method can evaluate quantitatively and accurately antiviral activity of lead compounds. Actually,
MTT assay had beenused to our study,including ofanti-influenza virus activity analysis for chlorogenin 3-O-β-chacotrioside derivatives, and their cytotoxicity evaluation. In addition, CPE method had also been usedin titration of influenza viruses and TCID50 assay in our study.
11.Materials and Methods: It is too lengthy and too detailed. Reference citations seem to be enough in most of the sections.
Response:
We really appreciate the reviewerfor kind recommendation. Meanwhile, thanks for the good suggestion. We again carried outediting carefully, and hope that the revised manuscripts can meet with approval.
12.Tables 1 and 3: The numbers should be summarized in same significant digits. Table 2: Quantitative measurement is required.Figures 3, 5, 7: Plaque titrations should be performed again to make them countable.
Response:
Thanks for the good suggestion.As suggested, we have combined table1 and 3 into same one. This table is shown as follows.
Table 1. Antiviral activities of the tested compounds against H5N1 viruses in MDCK cells(± S, n=3).
|
Compounds |
CC50(µM) |
IC50(µM) |
SI |
|
ribavirin |
132±0.87 |
37.20±1.6 |
3.55 |
|
UA-Nu-ph-5 |
127±1.697 |
15.59±2.4 |
8.15 |
|
XC-27-1 |
1247±1.16 |
16.83±1.45 |
74.1 |
|
XC-27-2 |
73.78±0.79 |
12.45±2.27 |
6 |
|
GA-NEG2-5 |
93±1.45 |
>100 |
NA |
|
WEW-35 |
76±0.79 |
>100 |
NA |
|
UA-Me-5 |
3.125±0.65 |
3.914±0.214 |
0.8 |
|
i-OA-Me-18 |
106±0.45 |
>100 |
NA |
|
S-100 |
68±0.729 |
>100 |
NA |
|
S-120 |
57±1.32 |
>100 |
NA |
|
S-130 |
100±1.817 |
>100 |
NA |
Notes: NA represents no activity.
In our study,the infectivity for HPAI H5N1virus was determined by TCID50 assay, which was calculated by Reed-Muenchmethod.What the reviewer said is right. TCID50 method can estimate the viral infectivity and virus content, but it cannot accurately determine the number of infectious virus particles. However, due to the poor reproducibility of PFU assay methods, now many studies apply TCID50 assay to determine infectivitytiter of viruses.
Table2.TCID50 titrations of H5N1subtype influenza virus
|
Number |
Virus dilutions |
|
||||||||
|
10-1 |
10-2 |
10-3 |
10-4 |
10-5 |
10-6 |
10-7 |
10-8 |
10-9 |
10-10 |
|
|
1 |
++++ |
++++ |
++++ |
++++ |
++++ |
++++ |
++++ |
- |
- |
- |
|
2 |
++++ |
++++ |
++++ |
++++ |
++++ |
- |
- |
- |
- |
- |
|
3 |
++++ |
++++ |
++++ |
++++ |
++++ |
++++ |
- |
- |
- |
- |
|
4 |
++++ |
++++ |
++++ |
++++ |
++++ |
- |
++++ |
- |
- |
- |
|
5 |
++++ |
++++ |
++++ |
++++ |
++++ |
++++ |
- |
- |
- |
- |
|
6 |
++++ |
++++ |
++++ |
++++ |
++++ |
- |
- |
++++ |
- |
- |
The calculation process is as follows:
|
Virus dilutions |
The number of wells |
Total number of wells |
CPEpercentage,% |
||
|
CPE |
Without CPE |
CPE |
Without CPE |
||
|
10-1 |
6 |
0 |
40 |
0 |
100% |
|
10-2 10-3 10-4 10-5 10-6 10-7 10-8 |
6 6 6 6 3 2 5 |
0 0 0 0 3 4 1 |
34 28 22 16 10 7 5 |
0 0 0 0 3 7 8 |
100% 100% 100% 100% 10/13(76.92%) 7/14(50%) 5/13(38.46%) |
|
10-9 |
0 |
6 |
0 |
14 |
0 |
|
10-10 |
0 |
6 |
0 |
20 |
0 |
Cytopathic effect (CPE):
Distance ratio=(76.92%-50%)÷(76.92%-38.46%)≈0.7
lgTCID50=0.7×(-1)+(-6)=-6.7
TCID50=10-6.7/0.1ml
PFUs=0.7×TCID50= 0.7×10-6.7/0.1ml=7×10-6.7/ml
All of the above, todilute the virus solution by 106.7 and inoculate 100µl, there are 50% of cells will occurcytopathic effect.
With regard to“plaque titration quantification”, actually, as descripted in the manuscript on Line 213-216, we had performed plaque formation reduction assay combined with serial dilution method, and this experiment was repeated at least three times.The results were shown in Fig.4, 6 and 8.
13.Tables 6 and 7, and Figure 10: It is recommended to combine.No data with 200micromolar concentration of the compounds against VSVG.
Response:
That’s really a great suggestion. As to the reviewer’s suggested, we combined Table 6 and 7 in the revised manuscript. As figure 10 is clear, the data of tables 6 and 7 are shown as supplementary data.
In this study,when the concentration was up to 100µM, the derivativespresented hardly any the inhibitory activity against VSVG pseudovirusby comparison with that of H5N1pseudovirus, so
we did not detect the inhibitory ratethe compounds against VSVGpseudovirusunder 200µM. To make our manuscript acceptable,we also have supplemented the experiment onthe antiviral inhibitory of the derivatives against VSVGpseudovirusunder 200µM. The inhibitory rate of XC-27-1 and XC-27-1against VSVGpseudovirusshowed 4.75±3.18% and 6.52±2.16% respectively.
Table 6.Inhibitionrate of derivatives against H5N1 and VSVG pseudoviruses
|
Concentration (µM) |
The inhibitory rate of derivatives (%) |
||||
|
XC-27-1 |
|
XC-27-2 |
|||
|
H5N1pseudovirus group |
VSVG pseudovirus group |
H5N1pseudovirusgroup |
H5N1pseudovirus group |
||
|
1.56 |
33.59±5.25 |
-0.74±6.43 |
26.83±9.96 |
2.62±4.57 |
|
|
3.13 |
38±3.24 |
-1.52±4.89 |
|
30.10±12.17 |
-3.39±4.64 |
|
6.25 |
40.69±10.88 |
-4.3±3.47 |
|
35.80±12.13 |
0.17±5.78 |
|
12.5 |
48.95±11.65 |
3.54±4.16 |
|
46.30±6.37 |
1.54±13.89 |
|
25 |
60.68±3.73 |
0.83±6.72 |
|
47.43±13.16 |
0.46±0.94 |
|
50 |
64.81±2.96 |
3.97±5.20 |
|
64.88±12.50 |
0.34±6.12 |
|
100 |
78.01±2.16 |
4.80±4.97 |
|
90.53±2.35 |
6.84±5.83 |
|
200 |
95.32±3.6 |
4.75±3.18 |
|
95.98±0.68 |
6.52±2.16 |
The inhibitionrateafter treatment withXC-27-1andXC-27-2 at two-fold serial diluted concentrations for 48h are means of three independent experiments (n=3, mean±S.E.M).Each group set 6 parallel wells.Differencewasconsidered statisticallysignificantwhen*p<0.05and**p<0.01vsVSVG pseudovirus group.
14.Fig 13, Figure legend: Fig11-b to Fig13-b.
Response:
Thank you so much for reminding us. We sincerely apologize for our carelessness. The“Fig11-b” under Fig 13was changed to “Fig13-b”on Line 755.
Finally, fully acknowledge all positive points from the reviewers.
Sincerely,
Ping Xiong
Associate Professor
Department of Pharmaceutical Engineering
South China Agricultural University
GuangZhou, China 510642
Phone: 0086-13533385789
Email: xp0000542003@scau.edu.cn ; 547273358@qq.com